# Augmented $CO_2$ tolerance by expressing a single $H^+$-pump enables microalgal valorization of industrial flue gas

Hong Il Choi [1,4], Sung-Won Hwang [1,4], Jongrae Kim [2], Byeonghyeok Park [3], EonSeon Jin [2], In-Geol Choi [3] & Sang Jun Sim [1✉]

Microalgae can accumulate various carbon-neutral products, but their real-world applications are hindered by their $CO_2$ susceptibility. Herein, the transcriptomic changes in a model microalga, *Chlamydomonas reinhardtii*, in a high-$CO_2$ milieu (20%) are evaluated. The primary toxicity mechanism consists of aberrantly low expression of plasma membrane $H^+$-ATPases (PMAs) accompanied by intracellular acidification. Our results demonstrate that the expression of a universally expressible PMA in wild-type strains makes them capable of not only thriving in acidity levels that they usually cannot survive but also exhibiting 3.2-fold increased photoautotrophic production against high $CO_2$ via maintenance of a higher cytoplasmic pH. A proof-of-concept experiment involving cultivation with toxic flue gas (13 vol% $CO_2$, 20 ppm $NO_X$, and 32 ppm $SO_X$) shows that the production of $CO_2$-based bioproducts by the strain is doubled compared with that by the wild-type, implying that this strategy potentially enables the microalgal valorization of $CO_2$ in industrial exhaust.

[1] Department of Chemical and Biological Engineering, Korea University, 145, Anam-ro, Seongbuk-gu, Seoul 02841, Republic of Korea. [2] Department of Life Science, Hanyang University, 206, Wangsimni-ro, Seongbuk-gu, Seoul 04763, Republic of Korea. [3] Department of Biotechnology, Korea University, 145, Anam-ro, Seongbuk-gu, Seoul 02841, Republic of Korea. [4] These authors contributed equally: Hong Il Choi, Sung-Won Hwang. ✉email: simsj@korea.ac.kr

Rapid increases in the atmospheric $CO_2$ concentrations have greatly impacted global ecosystems, and concomitant global warming, climate change, and ocean acidification severely threaten all life on Earth. To address the issues caused by increasing anthropogenic emissions, various carbon capture, utilization, and storage (CCUS) technologies have been developed for $CO_2$ reduction. Microalgae-based $CO_2$ conversion, *inter alia*, is considered one of the most promising and self-sustainable technologies because the relevant $CO_2$ sequestration process (i.e. photosynthesis) can be directly and entirely powered by solar energy and the level of productivity is significantly greater than that obtained with other biological technologies[1]. The wide range of product portfolios is another merit of this platform[2]. Despite their promising potential, the low tolerance of microalgae to high concentrations of $CO_2$ is one of the crucial thresholds that precludes the real-world application of microalgae for $CO_2$ removal[3]. In particular, this issue of $CO_2$ tolerance becomes more serious if phototrophic microalgae cultivation occurs in concert with an industrial exhaust gas stream, which is the most economically and energetically efficient biological process for removing $CO_2$ from air based on a life-cycle perspective[4] because the proportion of $CO_2$ in waste streams is commonly higher than 10%, a level that can significantly inhibit algal growth and unavoidably deteriorate the biological $CO_2$ conversion efficiency[5,6]. According to previous investigations, severe cytoplasmic acidification has been considered the principal reason for the impaired cell reproduction rate observed under high concentrations of acidic gas[7,8].

In recent decades, various efforts have been attempted to overcome the fatal weaknesses of microalgae. For instance, several microalgal species have been newly isolated from native environments, and studies have indicated that these species may be able to thrive under very high $CO_2$ conditions[9]. However, the selection process is time-consuming and laborious, and these algae hold low economic potential compared with other competitive strains due to insufficient information on the strain characteristics and a lack of species-tailored gene modification tools, which could block further strain improvements. Adaptive laboratory evolution approaches for improving the ability of algae to withstand high $CO_2$ environments through gradual acclimation have been performed and represent efficient methods, but the arduous and lengthy processes involved in this strategy are also problematic[10]. As an alternative, tolerance-enhancing trials based on genetic engineering represent a game-changing technology with a high potential to confer unprecedented improved tolerance, which can alleviate high $CO_2$ concentration-derived acid stress. However, limited successful cases have been reported, and the tolerance enhancement effect has only been verified under moderately high $CO_2$ conditions (5% $CO_2$), which is too low to directly address industrial exhaust gases[8].

Therefore, $CO_2$-tolerant microalgal cell lines and relevant strain improvement methods must be developed to expedite the realization and valorization of microalgae-based $CO_2$ reduction technologies and products. Here, we show a genetic engineering-based strategy to improve the $CO_2$ tolerance of microalgae that could be broadly applied to various species. Based on the results of the transcriptional profiling of a model microalga, *Chlamydomonas reinhardtii*, under extremely high $CO_2$ conditions, we identify the genetic principle of $CO_2$ tolerance. We demonstrate that the overexpression of a single PMA gene resulted in greater than two-fold increases in the algal biomass and lipid productivities by directly converting flue gas containing $CO_2$ and various acidic gases. We show that this strategy improves the microalgal $CO_2$ fixation performance, lowers the biomass production cost, and eventually contributes to the establishment of profitable microalgae-based $CO_2$ conversion technologies.

## Results

**Transcriptome landscaping during high $CO_2$ conditions.** We first performed an RNA sequencing (RNA-seq)-based transcriptome analysis of *C. reinhardtii* under extremely high $CO_2$ conditions (i.e. a 20% $CO_2$-enriched condition) to investigate its cellular response to a high $CO_2$ environment and identify a gene candidate that could augment $CO_2$ tolerance[11]. In addition, transcriptome profiling of algae grown in a $CO_2$-independent acidic environment (i.e. pH value of 5.0) was also carried out. The stress-imposing conditions were determined according to the results from preliminary condition screening experiments to determine the exposure levels of $CO_2$ and $H^+$ that reduce the algal growth rate by 50% (Supplementary Fig. 1). The RNA-seq analyses were then performed using six biological replicates of each condition (Supplementary Fig. 2). Because $H^+$ is known to play a key role in $CO_2$ poisoning under high $CO_2$ conditions and decrease the viability of algal cells (Fig. 1a), the latter analysis was conducted as a benchmark to potentially elaborate the $H^+$-related $CO_2$ tolerance mechanisms even though they are unlikely to share the exact same genetic response mechanism (Fig. 1b, Supplementary Fig. 3, and Supplementary Data 1)[7]. The assay of the specific growth rate under several high $CO_2$-mimicking conditions (i.e. conditions with $CO_2$-independent low pH and a high concentration of dissolved inorganic carbon (DIC); details are provided in Supplementary Note 1) indeed suggested the existence of subtle differences in the cellular responses to acidic conditions depending on the presence and absence of $CO_2$. The results showed that the decrease in the pH of the extracellular environment caused by high $CO_2$ levels could be a minor toxic factor, although it was not the only direct cause of the reduced cell viability. This finding strongly implies that the interaction between intracellularly absorbed $CO_2$ and cellular activity would aggravate intracellular pH ($pH_i$) acidification and thereby amplify the toxicity (Table 1 and Supplementary Fig. 4). Namely, $CO_2$ intoxication seemed to be synergistically caused by an influx of $H^+$ from the acidic extracellular space as well as the intracellular generation of $H^+$ via $CO_2$ hydration (facilitated by the activities of cellular enzymes, such as carbonic anhydrase; Supplementary Fig. 5). Notably, the toxicity of high [DIC], which could be accompanied by high $CO_2$ conditions and trigger dose-dependent stresses, such as osmotic pressure, was also evaluated, but the analysis revealed that it was not a toxic factor.

When exposed to high $CO_2$ levels, microalgae attempt to maintain their cytosolic pH, namely, pH homeostasis, against the environment by (i) upregulating proton extrusion extracellularly in response to the inward diffusion of $H^+$; (ii) inactivating the carbon concentrating mechanism (CCM) to minimize the collateral intracellular acidification caused by enriched $CO_2$ and facilitating $CO_2$ hydration intracellularly; (iii) modifying the composition of the cellular membrane to strengthen its role as a proton barrier; and iv) increasing ATP synthesis to provide biological energy for the operation of ATP-driven transporters and protein recovery processes[7,8,12,13]. Therefore, we probed the expression levels of several genes of interest related to the four abovementioned cellular responses based on a transcriptome analysis to examine whether these tolerance mechanisms function properly. In addition, the expression patterns of molecular chaperones (e.g. heat shock proteins; HSPs) were also inspected because these are ubiquitously expressed proteins under most stress conditions and play a crucial role in the recovery of proteins impaired by various stress factors[14]. In this context, the modulation of HSP expression is also an important response during acclimation to high $CO_2$-derived acidic conditions since the denaturation and aggregation of key proteins (i.e. enzymes) by excessively generated $H^+$ negatively contribute to cell proliferation[15]. Overall, 3867 and 1487 differentially expressed genes (DEGs; $log_2$fold change (FC) > 1, $p < 0.05$) were identified

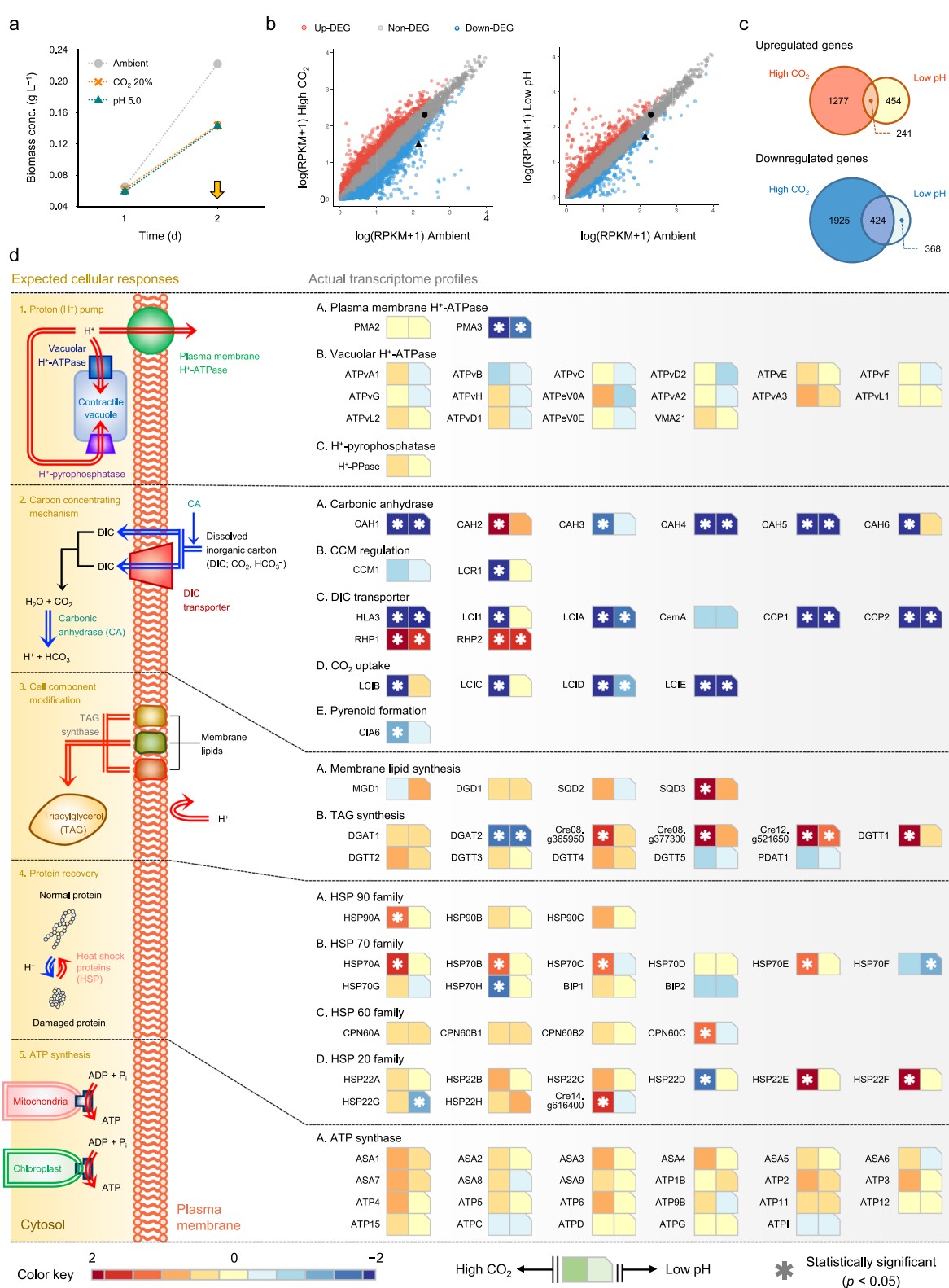

from the high $CO_2$ and low pH sets, respectively. Among these DEGs, 15.9% of the significantly upregulated and 18.1% of the downregulated genes in the high $CO_2$-treated cells were commonly found in the low pH-treated cells (Fig. 1c and Supplementary Data 1). These overlaps indicate that the algae may manage high $CO_2$-derived stress in a similar but not equal manner to counteract the stress attributed to low pH condition, and this conclusion is likely because the two inhibitory conditions may not exert identical toxic effects on the cell, as was experimentally revealed in this study (Table 1 and Supplementary Fig. 4).

Prior to the detailed inspection of individual gene expression levels, a Gene Ontology (GO) enrichment analysis was performed to extract and identify the biological processes (BPs) that are sensitively regulated in response to high $CO_2$ conditions. From

**Fig. 1 Transcriptome landscapes of the CO2 intolerant algal species *C. reinhardtii* (WT) exposed to extremely high CO2 (20%) and low pH (pH 5.0) conditions. a** Growth of the WT strain under high CO2 and low pH conditions. The conditions for EC50 were selected (Supplementary Fig. 1). The arrow indicates the time point of biomass sampling for RNA-seq ($n = 3$). Data are the mean ± SD of three biological replicates. **b** Scatter plot of the expression levels of each gene (in RPKM values). A pseudocount was added to all RPKM values to enable plotting. The transcriptional changes in the PMA2 and PMA3 genes are highlighted by black circles and triangles, respectively. **c** Venn diagrams of the number of genes that were commonly upregulated ($\log_2$Fold change (FC) > 1, $p < 0.05$) or downregulated ($\log_2$FC < 1, $p < 0.05$) under excess proton-derived stress conditions compared with their expression in cells grown under ambient conditions (as a control). The areas are to scale. **d** The left panel shows the predicted cellular responses to excess proton-derived stresses, and the right boxes present a heatmap of the $\log_2$FC values of the cell's knowledge-based selected genes. The detailed roles and descriptions of the genes under CO2-enriched conditions are described in Supplementary Data 3. An asterisk in the boxes indicates a statistically significant difference ($p < 0.05$) compared with the expression level under ambient conditions. An asterisk is shown only for the transcriptomes that showed over 2-fold changes in expression. In **c** and **d**, the statistically significant differences were determined by the generalized linear model (GLM) likelihood ratio (LR) test. Source data are provided as a Source Data file.

**Table 1 Comparison of the specific growth rate and concomitant pH of *C. reinhardtii* culture under high CO2-related stress application conditions.**

|  | Ambient (control) | High CO2 | Low pH | High [DIC] |
|---|---|---|---|---|
| μ (d⁻¹) | 1.253 ± 0.007 | 0.664 ± 0.004 | 1.162 ± 0.105 | 1.387 ± 0.005 |
| pH | 6.92 ± 0.04 | 6.23 ± 0.01 | 6.25 ± 0.05 | 7.32 ± 0.01 |

The data are shown as the average values ± standard deviations of the mean ($n = 2$). Data are mean ± SD of two biological replicates. DIC means dissolved inorganic carbon. The conditions for low pH (pH 6.2) and high [DIC] (7 mM) were determined by the corresponding [H⁺] and [DIC] values under high CO2 (20%) conditions, respectively. Source data are provided as a Source Data file.

the 55 enriched GO terms, diverse modes of cellular responses for enduring the high CO2 condition were determined (Supplementary Data 2). Among the upregulated gene clusters under the high CO2 condition, GO-BP terms associated with the maintenance of cellular homeostasis (GO:0019725, GO:0006873, GO:0030003, and GO:0055082) were enriched, which may be attributed to heightened cellular efforts for minimizing the radical change in the intracellular condition caused by the extracellular condition. The enrichment of GO-BPs relevant to carbohydrate metabolism (GO:0044262, GO:0016051, and GO:0034637) was likely a response to the increased cellular activities against the high concentration of CO2 because carbohydrates are known to not only help adjust the intracellular osmotic pressure but also act as protective agents for maintaining cellular homeostasis[16]. Additionally, under the high CO2 treatment, protein and amino acid metabolism-related GO-BP terms (GO:0006520, GO:1901605, GO:0009072, GO:0006551, GO:0009100, GO:0006528, GO:0030163, GO:0006511, GO:0019941, GO:0044257, and GO:0051603) were also found to be overrepresented, which was likely due to the activated cellular responses related to catabolizing proteins and amino acids that may lead to the production of NH3 and the reduction of intracellular H⁺ through deamination and decarboxylation, respectively, both of which are widely known as pivotal cellular mechanisms for microbial resistance to intracellular acidification[17].

Subsequently, the expression levels of the tolerance-related essential genes were assessed to verify the response behavior of the genes individually and to predict the detailed genetic cause of the low tolerance of *C. reinhardtii* to high concentrations of CO2 by comparing the purported cell responses with the actual gene expression patterns (Fig. 1d and Supplementary Data 3). As expected, a majority of essential genes that constitute the CCM were significantly downregulated (16 out of 21 investigated genes) under extremely high CO2 conditions as a counteraction to the increased environmental DIC concentration. Because an increased extracellular DIC concentration can trigger a series of reactions that ultimately results in excessive endomembrane acidification by the accelerated inward flux of DIC species and their intracellular hydration, which promotes additional generation and accumulation of H⁺ in the cytoplasm, this gene regulation seems to be an example of a cellular endeavor against

high CO2 conditions (Supplementary Fig. 5)[8]. Countertrends were found for CAH2, RHP1, and RHP2, and the plausibility of this hypothesis has been previously validated[18]. Interestingly, although the expression of CCM genes has been known to be mainly influenced by the external CO2 concentration, our findings showed that many CCM components (CAH1, CAH4, CAH5, HLA3, LCIA, CCP1, CCP2, LCID, and LCIE) expressed in response to low pH conditions showed similar patterns (i.e. downregulated) to those found under the high CO2 condition, which likely indicates that several CCM genes are also related to the response to CO2-independent low pH conditions. Regarding the determinants of cell components, the expression of key enzymes involved in controlling the composition of the cellular membrane appears to be regulated under extremely high CO2 conditions. For example, upregulated expression was observed for SQD3, which contributes to a decrease in the membrane fluidity of the cell envelope by increasing the composition ratio of saturated membrane lipids. In addition, the results showed increases in the expression levels of Cre08.g365950, Cre08.377300, Cre12.g521650, and DGTT1, which are responsible for converting unsaturated lipids in intracellular and plasma membrane (PM) regions into triacylglycerol (TAG), a typical energy storage sink synthesized under various stress conditions. Consequently, these responses may inhibit the entry of H⁺ into the cell, which decreases passive proton influx and eventually leads to stress evasion[19–22]. Additionally, as expected, under high CO2 conditions, various HSPs from a wide range of HSP subfamilies were highly expressed. These increases in the expression levels of stress-inducible molecular chaperones may contribute to the turnover and refolding of proteins that are denatured and aggregated by excessive H⁺-derived oxidative stress[13]. Meanwhile, a higher expression of a number of building blocks for ATP synthase was also observed under harsh conditions. In particular, mitochondrial ATP synthase-associated (ASA) proteins, such as ASA1, ASA3, and ASA4, which are involved in the dimerization of ATP synthase, showed 1.93-, 1.59-, and 1.63-fold upregulation (all $p$ values were less than 0.01), which in turn can result in increased ATP synthesis activity of the ATP synthase complex[23,24]. These changes may correspond to cellular activity to increase ATP synthesis because acidic conditions lead to the consumption of ATP for the

operation of $H^+$-pumps and the activation of ATP-dependent HSPs, among other processes[7,25].

Under extremely high external $CO_2$, the $CO_2$-intolerant species *C. reinhardtii* struggles to maintain its cytosolic neutrality by dynamically modulating genes associated with CCM, cell composition modification, protein recovery, and ATP synthesis in addition to previously elucidated $CO_2$ tolerance mechanisms. In contrast, the gene expression of $H^+$-pumps (e.g. PMAs, vacuolar $H^+$-ATPase – V-ATPase, and $H^+$-pyrophosphatase)[26] profiled under high $CO_2$ conditions reveals that $H^+$ sequestration from the cytoplasm to the extracellular region or contractile vacuole does not seem to be adequately regulated (i.e. not significantly upregulated) in the microalga, which is clearly different from previously reported expression trends observed with acid- and $CO_2$-tolerant microalgal species[7,11,13]. Rather intriguingly, a decrease in the expression of the PMAs – PMA2 (Cre10.g459200; slightly downregulated) and PMA3 (Cre03.g164600; highly downregulated with statistical significance) – was observed under high $CO_2$ conditions, which is an important finding because PMAs play a crucial role in life support during exposure to various acidic conditions[13,19]. The reproducibility of the gene expression results at the transcriptomic and translational levels under high $CO_2$ conditions was verified by qRT-PCR and immunoblotting (Supplementary Figs. 6a–c and Supplementary Table 1), respectively, which implies the mRNA-protein expression correspondence and, at the same time, suggests that the downregulation can directly influence the congenital low tolerance. Although their distinctive functions have not yet been explicated, PMA2 and PMA3 are two PMA proteins identified in *C. reinhardtii* with a sequence similarity of 47.4%, and both of these proteins have highly conserved regions that are generally found in well-characterized PMAs (Supplementary Fig. 7)[27–29].

Since PMA is a key and primary molecular workhorse located at the forefront of the cell and responsible for adjusting the intracellular $H^+$ concentration by pumping out $H^+$ into the extracellular region, which results in controlling the $pH_i$ and generating an electrochemical gradient to offer an $H^+$-mediated driving force for various secondary transport proteins[19,26], the atypical expression of PMAs under high $CO_2$ conditions may limit the discharge of excessive $H^+$ out of the cell, which would lead to undesired $H^+$ accumulation, even more acidification of the intracellular region, and disruption of the proton gradient for nutrient and ion transport. All of these changes can inhibit normal cell propagation (Fig. 1a and Table 1). Therefore, we conjectured that the $CO_2$ intolerance of the alga can be attributed to insufficient expression of PMA genes and the concomitant low ability of the cell to extrude the excessive $H^+$ generated under external $CO_2$ conditions. We thus hypothesized that the overexpression of PMAs would improve the $CO_2$ tolerance of the alga by increasing proton efflux and improving the ability of the alga to maintain its endomembrane pH at near neutrality even under high $CO_2$ environments.

**Construction of microalgal strains with high $CO_2$ tolerance**. To augment the $CO_2$ tolerance of *C. reinhardtii* by compensating for the low expression of $H^+$-ATPase, the overexpression target was a typical PMA, i.e. plasma membrane $H^+$-ATPase isoform 4 (hereafter referred to as PMA4) from *Nicotiana plumbaginifolia*, and its features have been well characterized and successfully expressed in various bioplatforms, such as other plants and yeasts[30,31]. If this pump can be activated in algal systems, the exogenous gene expression strategy can provide several merits. Because of its host-independent property, the expression of PMA4 holds the potential to be a versatile approach for various

microalgal species suffering from stagnant $H^+$ extrusion and consequent $CO_2$ intolerance, which is a general but critical issue that interferes with the practical application of microalgae-based $CO_2$ conversion technology[3,8]. In addition, PMA4 could help prevent cosuppression, which often occurs in homologous overexpression cases[32]. Given these findings, the gene insert was designed for mutagenesis. Because deletion of the autoinhibitory domain (located in the C-terminus; Supplementary Fig. 7), which is regulated by interacting with the regulatory 14-3-3 proteins that exist in every eukaryotic cell (including the microalga, *C. reinhardtii*), increases the activity of the $H^+$-pump[31,33], we decided to use the truncated form of PMA4 (referred to as PMA4ΔCter) instead of the full-length protein to confer the greatest acid-derived tolerance to the microalgal cell.

Subsequently, to label a reporter for probing the subcellular localization of the expressed acid-secreting nanomachine, the PMA4ΔCter coding gene was translationally fused at its 3′-coding-end to the monomeric Venus fluorescent protein (mVenus) coding gene with a flexible linker sequence (GGSGGGSG) (referred to as PMA4ΔCter-V) and placed under the control of a strong constitutive promoter (Hsp70A-Rbc S2) (Fig. 2a and Supplementary Fig. 8a). Afterwards, the codon-optimized insert cassette was delivered into the cell by electroporation.

After transformation, two independent mutants were screened — whose genetic and phenotypic stabilities are maintained over time – from selective medium plates containing Zeocin via colony PCR (Fig. 2b) using a designed screening primer set (Supplementary Table 1 and Supplementary Method 1). The transgenic cells were named PMA4ΔCter-V4 and PMA4ΔCter-V10 according to the order of their isolation. Due to the lack of an efficient homologous recombination method for *C. reinhardtii*[34], the inserted gene was randomly intercalated into the genome. Gene insertion leads to disruption of the original genome sequence, which can impact transgenic cell phenotypes. To clarify the cause of the phenotypic changes (*vide infra*), insertion site analysis was performed on the isolated strains by next-generation sequencing (NGS)-based whole-genome sequencing using HISAT2 software (Fig. 2c; see Supplementary Method 2 for details). In the PMA4ΔCter-V4 strain, only one insertion site was identified in chromosome 7, and this mutation deleted 23 base pairs (bps) from the genome and disrupted a predicted protein-encoding gene (Cre07.g331114). In PMA4ΔCter-V10, one inserted gene was also found on chromosome 6. During the insertion process, an approximately 5.5-kbp truncation of the genome occurred, and three protein-encoding genes, tail-specific protease 2 (TSP2; Cre06.g265850), a predicted protein (Cre06.g265900), and dynein heavy chain 3 (DHC3; Cre06.g265950), were unintentionally lost. The insertion site revealed by NGS was visually confirmed by PCR using appropriate primers (Supplementary Fig. 8b, c and Supplementary Table 1). Given the annotations of the disrupted genes (Supplementary Table 2) and previous reports on their functions[35–37], it is unlikely that the gene deletion events directly affected the ability of the transgenic strains to maintain their $pH_i$ against acid-derived $CO_2$ stress, which implies that the phenotypic modifications described below are generally not associated with the lost genes but rather caused by exogenous PMA expression. Following this analysis, the expression of the inserted gene was confirmed by quantitative real-time RT-PCR (Fig. 2d) using a designed primer set (Supplementary Table 1 and Supplementary Method 3), and the successful transcription of mRNA encoding PMA4ΔCter-V in the transgenic strains was confirmed by cDNA amplification. Moreover, no expression of PMA4ΔCter-V was detected in the wild-type (WT) strain. The expression of the protein of interest was then verified by Western blotting, and a band at approximately 130 kDa (Fig. 2e), which is

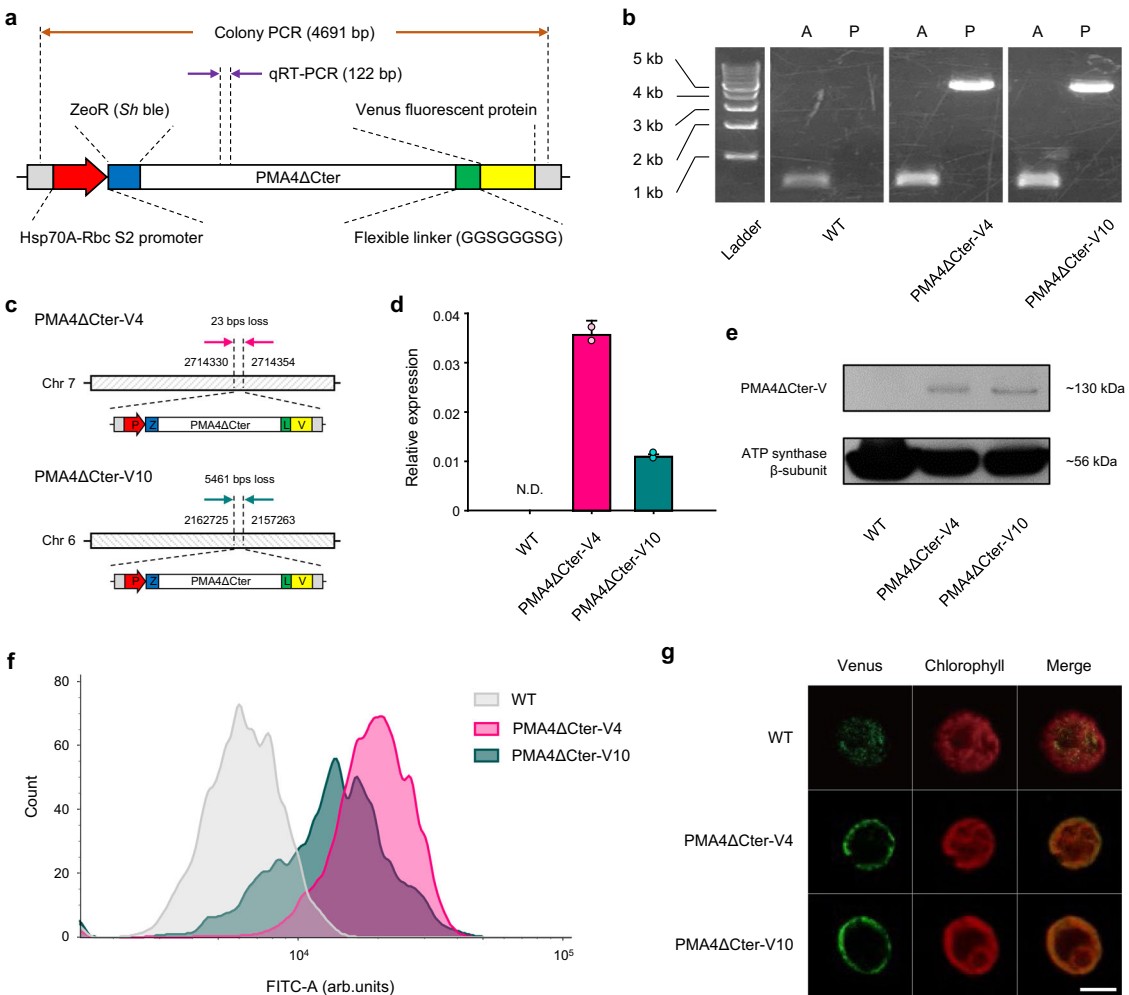

**Fig. 2 Mutant construction and confirmation. a** Simplified schematic representation of the codon-optimized PMA4ΔCter-V (PMA4 protein whose autoinhibitory domain in the C-terminus was deleted and fused with a fluorescent Venus tag) insert (total 6835 bp). The gene scheme is not to scale. **b** Genotyping of two Zeocin-resistant colonies by PCR to confirm insertion. A and P indicate the actin (an endogenous control) gene (expected amplicon length: 501 bp) and the PMA insert gene (expected amplicon length: 4691 bp), respectively. **c** Identification of the insertion site revealed by NGS. In the transgenic algal cells, a single insert was confirmed. In the cartoon, Chr, P, Z, L, and V represent chromosome, promoter, Zeocin resistance, linker, and Venus tag, respectively. The scheme is not to scale. **d** mRNA expression verification by qRT-PCR. Data are the mean ± SD of two biological replicates. The relative expression levels were normalized to the expression levels of a housekeeping gene (IDA5). N.D. stands for not detected. **e** Western blotting analysis for confirming protein expression. The expression of the ATP synthase β-subunit was simultaneously examined as an endogenous control. **f** Fluorescence analysis of the expressed protein by flow cytometry. Details of gating strategy used for the analysis is described in Supplementary Method 5 and Supplementary Fig. 9a–d. **g** Representative confocal images (bar scale of 5 μm) to confirm the proper localization of the exogenously expressed protein. In contrast to the results obtained for the WT strain, the fluorescence signals were observed exclusively in the plasma membrane of the transgenic strains. Source data are provided as a Source Data file.

located near the expected molecular weight of the PMA4ΔCter-V protein (120.81 kDa; estimated from its sequence), was visualized only with the transgenic strains when the same amount of protein is loaded (i.e. 20 μg), which reveals PMA overexpression properly occurred in the mutants (see Supplementary Method 4 for details)[31].

In addition to molecular-level expression tests, whether the protein is expressed in the intended subcellular part (i.e. PM) should be determined before evaluating the cellular performance because the protein must be localized appropriately before it can pump $H^+$ into extracellular space[31]. The localization of the protein was examined with the aid of a fluorescent tag protein (i.e. mVenus) (see Supplementary Method 5 for details). For precise fluorescence comparison, a transgenic microalgal cell line expressing only mVenus (referred to as V8) was constructed using a separate insert

cassette (Supplementary Fig. 8d) as a fluorescence reference strain. The integration of the transgene was genotypically verified (Supplementary Fig. 8e). First, the quantitative fluorescence of the cell lines was estimated by flow cytometry using BD Accuri™ C6 Plus software and Flowjo software (Fig. 2f). The results showed that mVenus expression induced significant shifts in the fluorescence peaks toward a higher intensity in the transgenic algal strains, including PMA4ΔCter-V4, PMA4ΔCter-V10, and V8 (Supplementary Table 3 and Supplementary Fig. 9e). Subsequently, the localization of the fluorescence source in *C. reinhardtii* was specified by confocal microscopy. The image-based assessment visually demonstrated that bright fluorescent proteins were preferentially concentrated near the PM of PMA4ΔCter-V4 and PMA4ΔCter-V10, whereas no notable concentration of fluorescence in the specific subcellular region was detected in the WT (Fig. 2g and

Supplementary Fig. 9f–i). This result indicated that the chimeric protein was successfully situated at the targeted position, namely, the PM, and ensures that the minimum requirement for proper protein function was met. In the fluorescent reference strain, V8 (Supplementary Fig. 9i), a strong fluorescence signal was observed throughout the cytoplasm, which suggested that the fluorescent proteins were floating freely without showing any specific localization. Compared with the fluorescent reference strain, the PMA-expressing cell lines exhibited relatively weak fluorescence, which can be primarily explained by the C-terminal truncation-derived lower stability and higher turnover of the expressed protein, as reported previously[31]. Another possible reason is that the protein expression level may not be that high because excessive protein expression can be harmful to the host, and in particular, excessive expression of PMA proteins can impose a huge burden on cells and consequently adversely affect their viability[38]. Either way, the pump protein was deemed to be sufficiently expressed at the posttranslational level to confer tolerance to $H^+$-related stresses without having a notable negative influence (*vide infra*). To further corroborate the localization of the protein, the fluorescent tag protein was probed by enzyme-linked immuno-sorbent assay (ELISA) using the PM-enriched fraction isolated with the two-phase partitioning method. Detection of the fluorescent tag in the sample also confirmed the proper localization of the protein (Supplementary Fig. 9j).

**Characterization of the performance of the transgenic cells**. Since the $CO_2$ tolerance improvement strategy described in this study is in line with the approach of enhancing the tolerance to low pH conditions, the productivity of the low pH-grown cell lines was evaluated first. The evaluation of the pH tolerance of the algal cells was conducted under two independent cultivation modes: mixotrophic and autotrophic conditions. For the mixotrophic culture of the facultative autotroph, acetic acid was applied as an organic carbon source. At a pH value of 5.5, the transformed cell lines showed a clear difference in growth performance compared with the WT cells. In particular, under mixotrophic cultivation at pH 5.5 (Fig. 3a, b), the proliferation rate of the WT cells was significantly stunted, and the productivity of the transgenic strains was high, which indicated that the acid tolerance of the mutants was successfully augmented. Because the growth of the transgenic strains at this low pH was comparable to that at neutral pH (Supplementary Figs. 10a and 11a), this constitutive PMA overexpression strategy did not appear to have a negative effect on the cells. In the photo-autotrophic acidic cultivation, the exogenous PMA-expressing mutants also exhibited a relatively improved growth performance under the same acidity (pH 5.5) but did not show a dramatic growth difference compared with the results obtained with the corresponding mixotrophic culture (Supplementary Figs. 10b, c and 11b, c). This trophism-dependent growth performance difference can be ascribed to the fact that the membrane-permeable weak acid protonated acetate (i.e. acetic acid), which was added to create the mixotrophic conditions, may exacerbate intracellular acidification by generating additional $H^+$ in the microalgal cells[38]. In this context, the acetic acid-supplemented pH 5.5 environment can inflict markedly higher levels of acid stress on the cells than a medium with an identical pH but without acetic acid, and this higher stress will eventually lead to the apparently dramatic difference in tolerance observed between the two different carbon regimes (Fig. 3b and Supplementary Fig. 11a–c). These results thus imply that the engineered cells possess high tolerance not only to extracellular acidic conditions but also to factors that aggravate $pH_i$ acidification.

Based on the previous pH tolerance evaluation, an autotrophic $CO_2$ tolerance test (without the organic carbon source) was performed with the cell lines after exposure to an extremely high $CO_2$ concentration (20%) to exclude the effect of other influencing factors on acid tolerance. Although the $CO_2$ concentration was somewhat higher than that found in typical industrial exhaust gas[6], it was selected to test whether the strains possessed sufficient acid tolerance to be used in practice given the additional acidification effect that may be induced by the various acidic gaseous components (e.g. $SO_X$ and $NO_X$) contained in general industrial exhaust gas streams. In this assay, the algal biomass production rates estimated from the turbidities of the cultures were compared to accurately assess $CO_2$ sequestration-related performance metrics because the microalgal $CO_2$ fixation rate ($R_{CO_2}$) is directly proportional to biomass productivity[2]. For the estimation, an $OD_{800}$-biomass correlation curve (Supplementary Fig. 12) and Eqs. (2) and (3) were used. According to the tolerance test, the WT strain showed stagnant biomass productivity (especially in the initial culture phase; Supplementary Fig. 11d) when exposed to high $CO_2$ concentrations compared with its growth under ambient conditions, which clearly reveals the vulnerability of these algal cells to elevated $CO_2$ levels. According to the entire batch culture result, eventually, the WT showed 0.66-fold decreased maximum biomass production while exhibiting 1.09-fold increased productivity under high $CO_2$ conditions (by Day 6) compared to those under ambient $CO_2$ conditions (by Day 17) during reaching their maximum cell growths (Fig. 3c and Supplementary Fig. 11c). The higher productivity is mainly attributed to the relatively shorter cultivation period because of the rapid cell death by high-$CO_2$ stresses. However, the slight increase would be hardly meaningful in terms of the $CO_2$ removal because the autotrophic growth rate that is not correspondingly increased against the elevated supply of $CO_2$ to the algal culture, inevitably causes the generation of unfixed $CO_2$, which significantly lowers the overall $CO_2$ reduction efficiency[39]. Moreover, the dramatically reduced maximum cell density would also be unfavourable from the same point of view, because the final biomass concentration is one of the most influential factors affecting the economic feasibility and energetic efficiency — which is intimately related to the net $CO_2$ removal ability of $CO_2$ conversion processes — from the life-cycle perspective[2,40].

On the contrary, the transgenic microalgal strains exhibited noticeable improvements in any photosynthetic production metrics, including maximum biomass productions and rates, even in a culture system to which an extremely high level of $CO_2$ was provided. Their maximum biomass productions were considerably above that found for the WT strains under identical $CO_2$ conditions, and 3.22- and 3.03-fold improvements were found for PMA4ΔCter-V4 and PMA4ΔCter-V10, respectively (upper panel of Fig. 3c). The accumulative productions and the rates of the transgenic strains under high $CO_2$ conditions (by Day 7) were also much higher (rather 1.12- and 1.09-fold improvements in the accumulative biomass productions and 2.77- and 2.62-fold improvements in the productivities for PMA4ΔCter-V4 and PMA4ΔCter-V10, respectively) than those under ambient $CO_2$ conditions (i.e. autotrophic pH 7.0) (by Day 17) (Fig. 3c and Supplementary Fig. 11c), which illustrates that the PMA-expressing mutants can even beneficially utilize excessive $CO_2$ while attenuating the intracellular acidification effect. This growth improvement also indicates that the amount of $CO_2$ fixation obtained with each mutant during the culture period was substantially improved compared with that found for WT under conditions with high $CO_2$ provision (lower panel of Fig. 3c). In terms of the rate of $CO_2$ fixation, PMA4ΔCter-V4 and PMA4ΔCter-V10 showed 3.23- and 2.99-fold increased $R_{CO_2}$

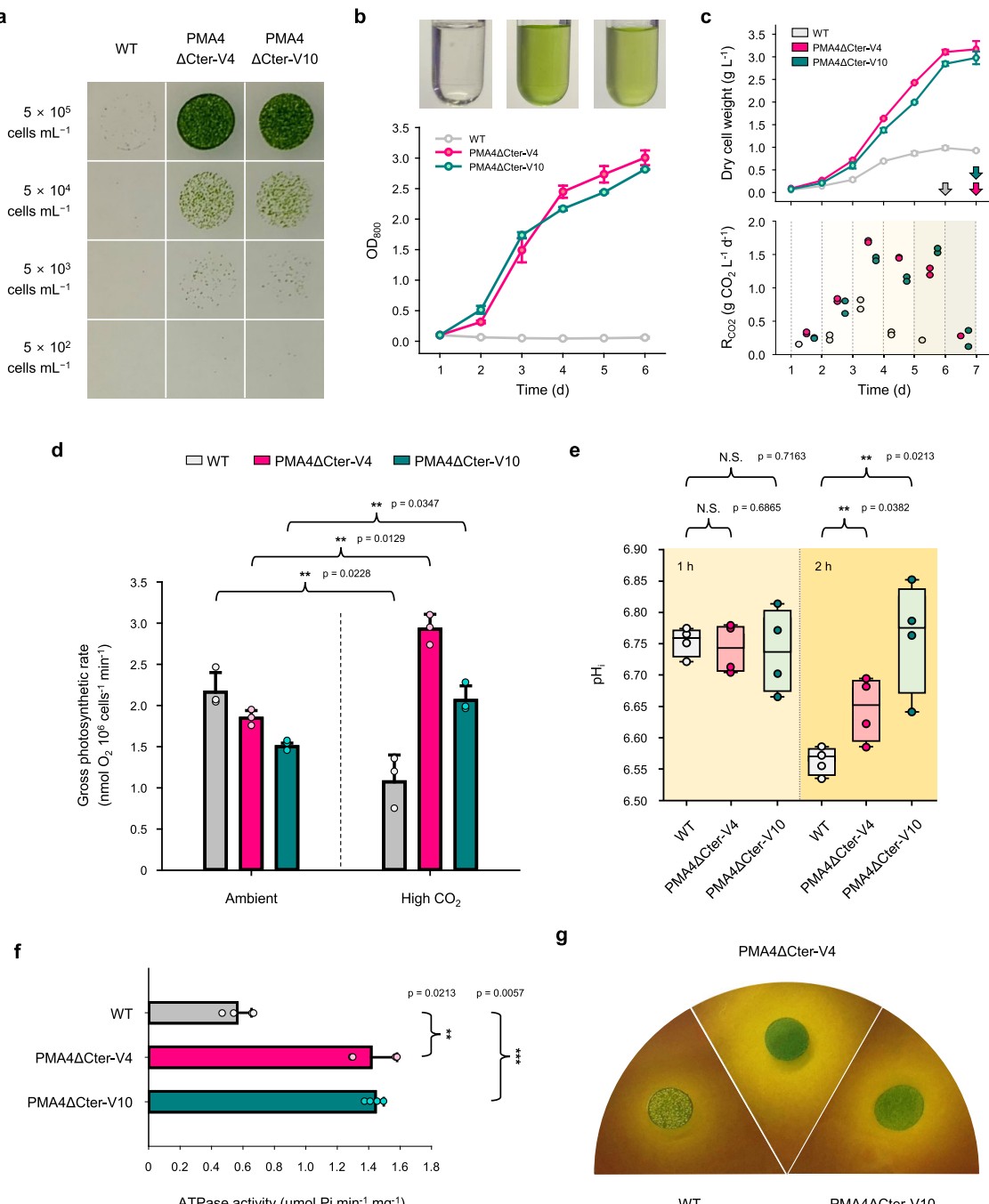

**Fig. 3 Cell performance of the transgenic strains compared with the WT strain. a** Spot culture with serial dilution on agar medium (supplemented with acetic acid) at pH 5.5. **b** The pH tolerance during liquid cultivation (pH 5.5; supplemented with acetic acid) is markedly enhanced via PMA4ΔCter expression. Shown are data from duplicate cell culture. The cell states at Day 2 of WT, PMA4ΔCter-V4, and PMA4ΔCter-V10 are displayed from the far left of the top panel. **c** The photoautotrophic biomass production (upper panel) and $CO_2$ fixation rate ($R_{CO2}$; lower panel) are improved under extremely high $CO_2$ conditions (20% $CO_2$). The two graphs share the same timescale. The arrows indicate the time points (distinguished by the colors) that the cell lines reached their maximum concentrations. Data from duplicate cell cultures are shown. Each dot in the lower panel depicts two biological replicates. **d** Per-cell gross photosynthetic rate of the strains exposed to high $CO_2$. The parameter was estimated as light-dependent $O_2$ evolution plus dark respiration. The double asterisks indicate statistically significant differences in the gross photosynthetic rate of each strain under the high $CO_2$ environment compared with those under the ambient conditions ($p < 0.05$). Each bar represents the mean ± SD of three independent experiments. **e** Box and whisker plots demonstrating the $pH_i$ of the microalgal strains exposed to high $CO_2$ conditions derived from four biological replicates. The median (the center line) ± whiskers (1.5× the interquartile range from the lower and upper quartiles) with the interquartile range (the boundaries) are provided. The double asterisks indicate statistically significant differences compared with the $pH_i$ of the WT strain after 2 h of exposure to $CO_2$ ($p < 0.05$). N.S. represents not significant. **f** In vitro ATPase activity assay using isolated PM-enriched fraction samples. The double ($p < 0.05$) and triple ($p < 0.01$) asterisks denote statistically significant differences compared with the ATPase activity of the WT strain, respectively. Each bar represents the mean ± SD of four independent experiments. In **d**, **e**, and **f**, the statistically significant differences were determined by a two-tailed Student's t test. **g** Representative image of bromocresol purple-mediated in vivo ATPase activity test. Source data are provided as a Source Data file.

values compared with the WT strain, respectively, during their exponential growth phases when their effective $CO_2$ fixations occur (between Day 2 and 6). Taken together, this tolerance improvement strategy raises the mutants' maximum biomass concentrations during a shorter culture period under high $CO_2$ supply conditions and improves their $CO_2$ removal rates and efficiencies, all of which are requisites of microalgal strains for efficient $CO_2$ sequestration and economically feasible $CO_2$-derived product productions[39,40].

In parallel to the growth assessment, the per-cell basis photosynthetic $O_2$ evolution of the cell lines exposed to high $CO_2$ conditions was evaluated (see Supplementary Method 6 for details). Consistent with the productivity data, the high $CO_2$-grown WT strain exhibited a lower gross photosynthetic rate (i.e. $O_2$ evolution plus dark respiration) than the WT strain cultivated under ambient conditions (approximately 50.4% decrease), which implies that the algal photosynthetic apparatus was severely impaired by the high $CO_2$ condition. In contrast, the PMA4ΔCter-V4 and PMA4ΔCter-V10 strains grown under high $CO_2$ conditions showed increased photosynthetic $O_2$ evolution rates in comparison to those found during growth under atmospheric conditions (approximately 58.6% and 37.3%, respectively) (Fig. 3d). Based on observations from previous reports and the results of this study, this excess $CO_2$-related photosystem impairment is likely the primary reason for the growth retardation observed in the presence of high levels of $CO_2$[8,41]. Moreover, the increased $O_2$ evolution observed with the transgenic cells is attributable to the activation of oxygenic photosynthesis-related enzymes under tolerably high $CO_2$ conditions to facilitate biological $CO_2$ assimilation, as has been well documented[42]. Thus, *C. reinhardtii*, which is a congenitally $CO_2$-intolerant microalga, became resistant to high $CO_2$ conditions via PMA expression and its activated core machinery for photosynthesis was thereby able to generate an increased amount of oxygen.

To probe whether PMA expression indeed improves photosynthetic $O_2$ evolution and biomass productivity by providing cells with the ability to maintain their pHi near neutrality, we next monitored the time-lapse cytoplasmic pH changes after high $CO_2$ treatment using a membrane-permeable pH-sensitive fluorescent dye, BCECF, AM (2′,7′-bis(2-carboxyethyl)-5 (6)-carboxyfluorescein, acetoxymethyl ester) (see Supplementary Method 7 for details)[26]. To eliminate the background signal, including the autofluorescence of the cells (for all strains) and the fluorescence of the fluorescent probe (i.e. mVenus for the transgenic algal lines), from the fluorescence of interest emitted by the pHi-sensitive dye, the pHi of each algal strain was calibrated independently (Supplementary Fig. 13). Consequently, the WT strain showed a dramatic decline in their pHi during 2 h of exposure to high $CO_2$ (a median pHi value of 6.56), whereas transgenic cell lines showed a better ability to robustly maintain a near-neutral pHi under acidic conditions (median pHi values of 6.65 and 6.78 were obtained for PMA4ΔCter-V4 and PMA4ΔCter-V10, respectively) (Fig. 3e), which demonstrates that high $CO_2$ conditions largely affect the viability of microalgae by lowering the cytosolic pH and that PMA expression in algae results in a positive response to $CO_2$-derived intracellular acidification. Given that the photosynthetic $CO_2$ fixation system and $O_2$ evolution machinery are highly sensitive to the pH and $CO_2$ conditions of the surroundings as well as to the cytosolic and subcellular pH[43,44], the differences in the ability of the cells to maintain their pHi under high $CO_2$ conditions appear to be the fundamental cause of the tolerance gap between the WT and transgenic strains.

In the final stage of the cell performance characterization, we investigated the $H^+$ pumping ability of the algal cells determined by the activity of ATPase on the PM region to determine the $CO_2$ tolerance conferred to the mutants by exogenously expressed PMA via the $H^+$ extrusion function of the cell. This ability was comprehensively evaluated both in vitro and in vivo. In the in vitro ATPase activity test, the PM fraction-enriched protein sample, which is identical to that isolated for assessment of its localization (i.e. ELISA), was used (see Supplementary Method 8 for details)[31]. As expected from the results of the previous performance tests, PMA4ΔCter-V4 and PMA4ΔCter-V10 exhibited approximately 2.5- and 2.6-fold higher ATPase-mediated ATP hydrolysis performance than the WT strain, respectively, which demonstrates that the transgenic strains have substantially higher $H^+$ translocation activity in their PM regions than the WT strain (Fig. 3f). The greater ATPase activity in the mutants could cause higher ATP consumption, which may lead to additional energetic costs associated with the exogenous ATPase. However, given algal growth under neutral conditions (Fig. 3b and Supplementary Fig. 11a, c), the viability of the algal cells was not significantly affected. An in vivo assay was performed to confirm the algal $H^+$ extrusion performance at the cell level and the direction of the $H^+$ flux using a chemical pH indicator, bromocresol purple (see Supplementary Method 9 for details). The colourimetric assay visually demonstrated that the $H^+$ flux was appreciably improved in the PMA-expressing mutants, as can be inferred from the color change on the perimeter of the cells caused by the expelled $H^+$ and the accompanying extracellular acidification (Fig. 3g and Supplementary Fig. 14), and this result also provides direct evidence of the outward flux direction of $H^+$, namely, from the intracellular region to the exofacial space, and hints that the $H^+$ pumping protein was properly localized and oriented as desired. On balance, the data clearly show that the expression of the $H^+$-pump imbued the algal transformants with markedly improved $H^+$ efflux and an augmented ability to maintain their pHi against factors causing severe intracellular acidification (e.g. membrane-permeable organic acids and $CO_2$); thus, the algae showed enhanced tolerance to extremely high $CO_2$ conditions.

**Directly utilizing coal-fired flue gas for fuel production.** To estimate the real-world practicality of the engineered algal strain as a $CO_2$-derived biofuel production platform[45], we grew the PMA4ΔCter-V4 strain, which showed remarkable productivity under the high $CO_2$ level (Fig. 3c), in the presence of actual coal-fired flue gas discharged from the Taean thermal power station complex unit 5 boiler of Korea Western Power Co., Ltd. The gas stream consisted of approximately 13% $CO_2$, 20 ppm $NO_X$, and 32 ppm $SO_X$ on average (vol.%)[46]. In this outdoor cultivation, coal-fired flue gas was chosen as the exemplary industrial flue gas because the discharge of $CO_2$ from the exhaust is the largest contributor to global emissions, and its reduction is therefore vital[47]. Despite the urgency of mitigating this pollutant, the direct application of microalgae has been considered impractical to date due to its vulnerability to $CO_2$[41].

The strain was photoautotrophically cultivated with continuously provided $CO_2$ through the introduction of highly acidic exhaust gas as the sole carbon source in a microalgal $CO_2$ bioconversion facility constructed next to the coal combustion boiler using a 10 L (in working volume) photobioreactor (PBR) (Fig. 4a). After two weeks of cell culture, we holistically evaluated various performance indices of the strain using Eqs. (2)–(6)[45]. We observed that PMA4ΔCter-V4 exhibited a markedly improved phototrophic growth rate than the WT strain (2.28-fold higher) and beneficially converted the high concentration of $CO_2$ included in the noxious gas stream. The increased biomass productivity in the transgenic algal strain led to the enhancement

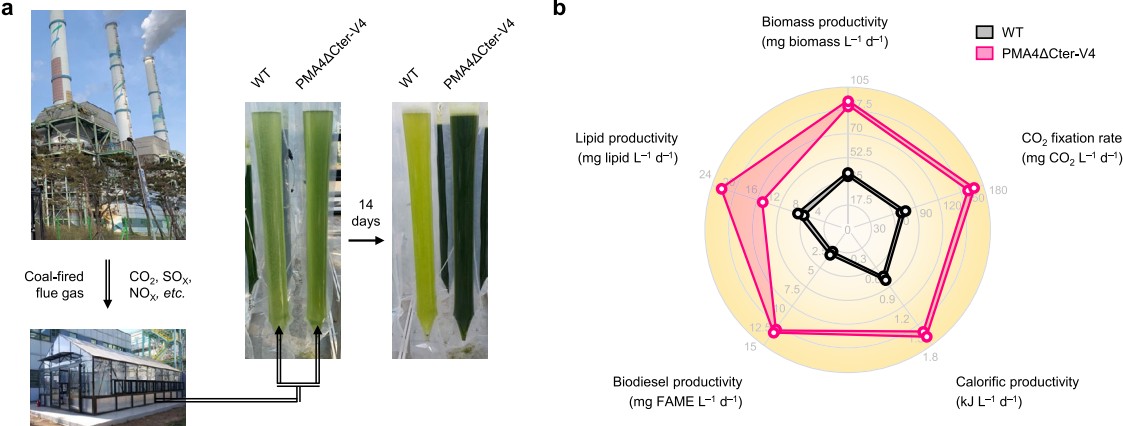

**Fig. 4 Proof-of-concept biomass and biofuel production with the engineered strain PMA4ΔCter-V4 using coal-fired flue gas. a** Bioconversion of $CO_2$ in the flue gas stream emitted from the coal-fired power plant. A house-manufactured bubble column PBR (10 L of working volume) was employed for cultivation in the vicinity of the coal combustion boiler. **b** Radar plot showing various cellular performances measured in duplicate ($n = 2$). The covered ranges on the axis indicate the standard deviation of each parameter. Source data are provided as a Source Data file.

of other essential parameters, including the microalgal $CO_2$ fixation rate (by 2.23-fold), total lipid productivity (by 2.21-fold), total fatty acid methyl ester (FAME) productivity (by 4.68-fold), and calorific productivity (by 2.20-fold) (Fig. 4b, Supplementary Fig. 15, and Supplementary Table 4). The acceleration of $CO_2$ sequestration, FAME accumulation, and calorific production indicate the practicality of using the engineered strain as a biological platform for producing various types of biofuels (i.e. biodiesel and solid fuel for direct combustion) while reducing industrial $CO_2$ emissions even when a variety of toxic gaseous compounds are continuously delivered into the culture. Overall, PMA expression allows inherently $CO_2$-intolerant algae to overcome the cytotoxicity of high $CO_2$, which leads to the efficient and rapid conversion of greenhouse gases and improves the technological feasibility of using microalgae with augmented tolerance.

## Discussion

To date, numerous studies have attempted to address the low $CO_2$ tolerance issue, which is a key hurdle that reduces the applicability of microalgal $CO_2$ reduction technology. Due to the lack of comprehensive analyses of the mechanisms underlying the intolerance of microalgae, previous approaches only focused on resolving the incidental causes of vulnerability; thus, the resulting improvements did not meet the requirements for practical applications with various types of industrial flue gas[8]. Therefore, we thoroughly inspected the global gene expression patterns of microalgae in response to extremely high $CO_2$ conditions using the model microalga *C. reinhardtii* to specify the immediate reason for its susceptibility to acidic gas. The investigation revealed that the representative intolerant species exhibited a relatively low expression level of the core $H^+$ pumping system, which is inconsistent with the previously reported response mechanisms to high $CO_2$-derived stress observed in $CO_2$-tolerant microalgae[7,11,13]. Notably, PMAs, which regulate the $pH_i$, were rather downregulated (PMA2 was slightly downregulated and PMA3 was significantly downregulated); thus, we postulated that aberrant expression of the $H^+$-pumps is the main factor for the ongoing intracellular acidification, which ultimately leads to a severe decline in cell viability. The reason for the unpredicted gene expression levels may not be clearly explained only by the investigations performed in this study, but one speculation —

regardless of the exterior acidity (i.e. both high $CO_2$ and $CO_2$-independent low pH conditions impose acidic stress on the cell), the expression of the PMA genes in the $CO_2$-intolerant alga is always negatively regulated by the abundance of extracellular $CO_2$ as if the genes are constituents of CCM – could provide evidence of the condition-dependent gene expression patterns detected in this study (Supplementary Fig. 6a, b). A possible dependence of PMA expressions on the extracellular $CO_2$ concentrations was also suggested by a recent comprehensive proteomic study of *C. reinhardtii*[48]. The results showed that PMAs (both PMA2 and PMA3) of the alga may be subordinate to the CCM and intimately interact with its core elements (e.g. HLA3 and LCI1). Since the gene regulation of CCM and its components is heavily influenced by the external $CO_2$ conditions, the functional relevance exhibits the possibility of an effect of exterior $CO_2$ on PMA expression. However, additional studies are needed to precisely conclude the association of the two machineries (i.e. the $H^+$-pump and carbon concentrating system). In parallel, characterization of the distinctive biomolecular properties of algal PMAs, such as their putative 14-3-3 protein-independent auto-inhibitory mechanisms, merits further study to provide a better understanding of the proteins and furthermore the organism, which would ultimately facilitate the greatest use of the microorganism's potential abilities (Supplementary Fig. 7)[28,29,49].

As predicted from the transcriptomic analysis, we introduced a typical PM-bound $H^+$ transporter, whose expression is initiated by a strong constitutive promoter to enable continuous PMA expression with the low possibility of intervention by the host's gene expression controlling mechanisms (e.g. CCM, *vide supra*), into congenital $CO_2$-susceptible microalgae to ameliorate the intolerance problem and rigorously verified the characteristics of the generated transgenic microalgae. The WT strain showed a reduced autotrophic growth rate under high $CO_2$ conditions compared with that under ambient conditions, which may be attributed to its high $CO_2$-labile $pH_i$ property and the resulting damage to its photosynthetic activity. In contrast, the $CO_2$ tolerance of both mutant strains was clearly augmented, which resulted in markedly increased biomass productivity even under harsh conditions by facilitating $H^+$ efflux, thereby reinforcing the $pH_i$ maintenance ability and contributing to the activation of oxygenic photosynthetic apparatuses in combination with sufficient DIC from the surroundings[42]. In the proof-of-concept experiment, microalgal biofuel production using a coal-fired flue

gas with a high level of $CO_2$ was investigated, and the engineered algal strain exhibited over twofold increases in $CO_2$-originated biomass and biofuel productivities without any process intervention, which may negate the reduction in $CO_2$ efficiency of the overall system[2]. These findings demonstrate the applicability of microalgae as a practical platform for the conversion of $CO_2$ in the flue gas to produce diverse sustainable products. Strain improvement technology can mediate the utilization of highly cytotoxic waste streams as a fully serviceable carbon source for microalgal cultivation and will thus aid the use of microalgae as a biological platform for practical $CO_2$ reduction.

In addition to the environmental benefit, the enhanced tolerance and resulting improvements in biomass and lipid productivity can contribute to promoting the economic feasibility of microalgal systems, including the realization of the price of biofuel. According to previous studies, 2.28- and 4.68-fold increases in biomass and biodiesel-convertible lipid (i.e. FAME) productivities could result in 58.6% and 74.3% decreases in the biodiesel cost, respectively. These decreases can be approximated from the estimation that the productivities and resulting biofuel costs are anticipated to decline by exponential factors of 1.07 and 0.88, respectively[50]. The upgraded productivities will undoubtedly serve as a catalyst for lowering the production cost of microalgal biofuel in various forms, expanding the market base of biofuel, and ultimately valorizing the overall microalgal industries. The need to use closed cultivation systems (i.e. PBRs) for biocontainment to prevent undesired dissemination of genetically modified (GM) microalgae to the environment and the resulting increase in production costs can be a concern. Even though the cultivation of GM algae in open environment does not seem to be accompanied by greater environmental risks than expected, public concerns are not negligible[51]. Therefore, PBR systems should be inevitably exploited for the cultivation of GM algae. In fact, according to an extensive techno-economic analysis study of microalgal cultivation, PBRs can be used as equipment for moderating the price of products in the near future with ongoing technical developments. Their use obviously ensures better photosynthetic efficiency and shorter light paths, which are key parameters lowering the production costs of biomass. These properties eventually contribute to improving the economic feasibility of the process by counterweighting the costs related to the installation and operation of PBR[52]. In addition, considering the goal of the cultivation — practical $CO_2$ reduction with phototrophic microalgal culture — PBRs constitute the most suitable platform for serving the purpose of enabling intensive bioconversion of $CO_2$. Thus, harnessing well-designed and cost-effective PBR systems may simultaneously guarantee thorough containment of GM organisms, efficient $CO_2$ fixation, and high biomass productivity while securing economic competitiveness.

We found that the expression of exogenous PMA is beneficial for algae because it significantly improves their tolerance to acidic conditions (i.e. low pH and high $CO_2$) and does not compensate for growth under neutral pH conditions. The simple structure of PMAs — i.e. the protein comprises a single polypeptide — seems to play a pivotal role in successful PMA expression and concomitant tolerance enhancement without burdening the transgenic cells[53]. By virtue of this structural property, it is likely that the desired augmentation was directly achieved by overexpression of a single protein unit. In contrast, straightforward tolerance enhancement would have been quite challenging if the overexpression target was a gene encoding a subunit of a multisubunit $H^+$ pumping protein such as V-ATPases. Since such proteins are congenitally composed of several subunits and normally function only when the complex is well organized with the proper quality and quantity of subunits, a possible approach that involves the overexpression of several specific subunits would lead to protein malfunction due to abnormal subunit assembly and ultimately thwart the improvement attempt[54]. A strategy that aims at overexpressing an entire protein complex (i.e. individual overexpression of an entire subunit) could also be problematic because multigene overexpression may impose an enormous genetic burden on the transgenic cell line. Despite the expected difficulty and intricacy, uncovering the V-ATPase function in $CO_2$ tolerance remains an interesting topic for future work because V-ATPases are also known to play an important role in the maintenance of the $pH_i$ against extracellular acidic conditions[26].

Another advantage of this strategy is that the single $H^+$-pump can be universally expressible in various organisms and causes a dramatic increase in the algal $CO_2$ tolerance. Thus, this simple but highly effective improvement strategy is likely applicable to multiple algae and not limited to a particular species. In particular, the ability to enhance robustness will be greatly effective for programmable microalgal chassis because potential cell factories can be easily transformed to photoautotrophically produce various $CO_2$-derived sustainable products by additional transformation but are generally unable to tolerate environmental stresses at practical levels[5,8,34,55]. On the one hand, this acid tolerance augmentation strategy could also be conjugated as a genetic approach for the maintenance of axenic conditions in acidic microalgal cultivation due to the considerably improved resistance toward an extracellular acidic pH[56]. Taken together, the results strongly provide a plausible and versatile solution for overcoming vulnerability to $CO_2$, a fundamental hurdle of microalgae-based $CO_2$ conversion technology, and thereby expedite the realization of $CO_2$ reductions systems and the commercialization of microalgal-derived products.

## Methods

**Microalgal strain, media formulation, and culture conditions.** *C. reinhardtii* WT strain CC-125 was obtained from the Chlamydomonas Resource Center at the University of Minnesota. All the engineered cell lines used in this study were constructed from this background strain. Laboratory-scale culture of the algae was conducted with modified Tris-acetate-phosphate (TAP; hereafter referred to as MTAP) and Tris-phosphate (TP; hereafter referred to as MTP) media, and the buffer capacities of the media were enhanced with an additional buffer material, namely, MES, to strictly maintain the acidity of the culture. Otherwise, the pH would be seriously lowered due to ammonium uptake, which can prevent accurate evaluations of $CO_2$ tolerance (Supplementary Fig. 4)[5]. The mixotrophic MTAP medium at neutral pH consisted of 10 mM MES, 17.4 mM acetic acid, 7 mM $NH_4Cl$, 0.83 mM $MgSO_4 \cdot 7H_2O$, 0.45 mM $CaCl_2 \cdot 2H_2O$, 1.65 mM $K_2HPO_4$, 1.05 mM $KH_2PO_4$, 0.134 mM $Na_2EDTA \cdot 2H_2O$, 0.136 mM $ZnSO_4 \cdot 7H_2O$, 0.184 mM $H_3BO_3$, 40 μM $MnCl_2 \cdot 4H_2O$, 32.9 μM $FeSO_4 \cdot 7H_2O$, 12.3 μM $CoCl_2 \cdot 6H_2O$, 10 μM $CuSO_4 \cdot 5H_2O$, and 4.44 μM $(NH_4)_6MoO_3$, and the pH was adjusted to a value of 7.0 with 25 mM Tris base[57]. All chemicals were purchased from Sigma-Aldrich (USA). Moreover, an autotrophic MTP medium (pH 7.0) was prepared using a method identical to that used for the MTAP formulation, although 0.35‰ HCl was added instead of acetic acid. The cells were grown at 23 °C in shaking incubators at a rotation rate of 120 rpm under continuous light illumination (50 μE $m^{-2}$ $s^{-1}$). The headspaces were provided with air and 20% $CO_2$-enriched air according to the purpose of each experiment. Cell growth was monitored daily using a UV-1800 UV spectrophotometer at 800 nm (Shimadzu, Japan).

To search for the factor underlying high $CO_2$ toxicity, media with $CO_2$-independent low pH (pH 6.2) and high [DIC] (7 mM) were prepared based on MTP medium. The pH was modified by the addition of HCl, and the [DIC] was adjusted by dissolving sodium bicarbonate. A [DIC] of 7 mM was selected based on a previous study that described the effect of the gaseous $CO_2$ concentration on the [DIC] in the liquid phase[5]. Moreover, a nonbuffered medium was used, and this medium was prepared using the same method as that used for MTP except for the addition of buffer materials (i.e. MES and Tris base). The pH was then autonomously adjusted to a value of 7.0. The specific growth rate of microalgae (μ; $d^{-1}$) was evaluated according to the following Eq. (1):

$$\mu = (lnOD_{800,F} - lnOD_{800,I})/(t_F - t_I) \qquad (1)$$

where $OD_{800,F}$ and $OD_{800,I}$ are the final and initial optical densities at 800 nm measured at two specific time points, namely, $t_F$ and $t_I$, respectively.

**RNA preparation and RNA-seq analysis**. To identify high $CO_2$- and low pH-sensitive genes, RNA samples from *C. reinhardtii* grown under specific $CO_2$ (20% $CO_2$) and pH (pH 5.0) conditions that suppressed the rate of algal growth by 50% compared with that observed under ambient control conditions (atmospheric pH 7.0) (the effective concentrations that caused a decrease in the growth rate; $EC_{50}$)[14] were isolated (Fig. 1a). Prior to culture, the $EC_{50}$ values of $CO_2$ and pH were predetermined (Supplementary Fig. 1). After one day of cell growth — the exposure time was determined as a short period during which the stress stimuli provoke stress-induced growth inhibitions at the target levels, before the cells enter in the center of their temporal adaptive stages (between Day 3 and 4 for the high $CO_2$-grown cells; Supplementary Fig. 1)[58] the algal cells were harvested by centrifugation ($1373 \times g$, 5 min) and immediately frozen using liquid nitrogen. The same cell samples were used in the immunoblotting analysis for investigating condition-dependent PMA expression patterns at the protein level. Subsequently, RNA extraction was performed using the PureHelix™ Total RNA Purification Kit (Nanohelix, Republic of Korea).

RNA-seq-based transcriptome analysis was performed using six biological replicates from each condition (Supplementary Fig. 2)[13,59]. To verify the quantity and quality of the RNA samples, both a NanoDrop™ spectrophotometer device (Thermo Fisher Scientific, USA) and an Agilent 2100 Bioanalyzer (Agilent Technologies, USA) were used. RNA samples showing an RNA integrity number (RIN) of approximately 6.0 were selected for library preparation using the TruSeq RNA Library Preparation Kit (Illumina, USA). RNA-seq was then conducted using the Illumina NovaSeq™ 6000 sequencing system, and 150-bp paired-end reads were generated. The resulting data sets were mapped to the reference genome of *C. reinhardtii* (*Chlamydomonas reinhardtii* v5.6, JGI) using Bowtie2 software with the default parameters. The Bioconductor R software package edgeR was employed to calculate the reads per kilobase per million mapped reads (RPKM) units of each gene and identify DEGs. Genes with $|\log_2 FC| > 1$ and $p < 0.05$ were considered significant DEGs. The statistically significant differences were estimated based on the GLM LR test. Using the DEG gene sets, the R package *topGO* was used to perform GO enrichment analysis, and Fisher's exact test method was used to determine the statistical significance ($p < 0.05$). To assign the GO terms, the relevant genes were annotated to Clusters of Orthologous Groups (COG) categories using the Kyoto Encyclopedia of Genes and Genomes (KEGG) database. Individual genes related to $CO_2$ tolerance (Fig. 1d and Supplementary Data 2) were selected and analyzed based on descriptions in the published literature[13,18,20–24,26,60–69] and gene annotations presented in the *Chlamydomonas reinhardtii* v5.6 database. Then, their expression patterns were profiled to comprehensively assess the effect of high $CO_2$ conditions on the cell at the transcriptomic level.

**Generation of the PMA-overexpressing line**. The coding sequence of PMA4, which is a well-studied PMA, from *N. plumbaginifolia* was obtained from the National Center for Biotechnology Information (NCBI, USA) database. The coding region of a C-terminal autoinhibitory domain within the original sequence (100 amino acids, 300 bp) that may reduce $H^+$ efflux was deleted to confer an improved $H^+$ extrusion ability to the engineered algal strain[30,31,33]. In addition to the truncated sequence, a green fluorescent protein (GFP) derivative, mVenus, was attached to PMA4ΔCter by a Gly-rich GGSGGGSG flexible linker to probe the localization of the exogenous protein, and the sequence of the final expression target PMA4ΔCter-V was eventually identified (Fig. 3a). The chimeric protein-coding sequence was then codon-optimized using GeneArt™ GeneOptimizer™ software (Thermo Fisher Scientific, USA) considering the codon usage preference in *C. reinhardtii* nuclei and synthesized using GeneArt (Supplementary Figs. 8f–h). Subsequently, the gene cassette was digested with *Eco*RI and *Bgl*II and cloned into the pChlamy_4 protein expression vector (Thermo Fisher Scientific, USA) containing a constitutive promoter (Hsp70A-Rbc S2) and Zeocin resistance gene (*Sh* Ble). The PMA4ΔCter-V-bearing plasmid was then linearized by *Sca*I prior to transformation. All restriction enzymes and T4 DNA ligase used in the cloning step were purchased from Thermo Fisher Scientific (USA). To fabricate the mVenus coding insert (Supplementary Fig. 8d), the corresponding sequence (i.e. PMA4ΔCter-V without PMA4ΔCter and the linker sequence) was cloned into the same expression vector backbone.

The engineered *C. reinhardtii* strain was transformed by electroporation according to the following protocol. The seed was freshly maintained and grown in 200 mL of conventional mixotrophic (TAP) medium until the cell concentration reached ~$1.5 \times 10^6$ cells mL$^{-1}$[57]. The cells were then collected by gentle centrifugation at $1373 \times g$ for 5 min and resuspended in 10 mL of MAX Efficiency™ Transformation Reagent for Algae (Thermo Fisher Scientific, USA). The washing step was repeated twice. The treated cells were harvested and resuspended in 1 mL of the same reagent. Four micrograms of the linearized insert were supplemented with 250 μL of the cell suspension and chilled at 4 °C for 5 min. The cell-cassette mixture was placed into a prechilled electroporation cuvette with a 2-mm gap. An exponential pulse was applied to introduce the exogenous gene into the algal cell using the BTX Electroporation System GEMINI X² (Harvard Apparatus, USA) with the following conditions: voltage, 500 V; capacity, 50 μF; and resistance, 800 Ω. Following electropermeabilization, the cells were incubated in the dark for 15 min and transferred to 10 mL of TAP medium supplemented with 40 mM sucrose. After 16 h of incubation in the dark, the cells were collected and

placed on a selective solid medium consisting of TAP medium, 1.5% (w/v) plant agar (Duchefa Biochemie, The Netherlands), and 5 μg mL$^{-1}$ Zeocin (Thermo Fisher Scientific, USA). The cells on the plate were grown for 7 days at 23 °C under low light conditions (50 μE m$^{-2}$ s$^{-1}$).

**Immunoblotting**. The primary antibodies used were $H^+$ ATPase (1:1,000; AS07 260) and AtpB (1:10,000; AS05 085) from Agrisera. The secondary antibody used was HRP-conjugated goat anti-rabbit IgG (1:100,000 for quantification of the condition-dependent PMA expression and 1:1500 for detection of the exogenous PMA protein; AS09 602) from Agrisera. The dilution rate of the secondary antibody was adjusted according to the manufacturer protocol of the developing reagents. In quantitative analysis of protein expression levels, the band intensity was measured using ImageJ software. Detailed experimental procedures are supplied in Supplementary Method 4.

**Primers**. A complete list of all primers used, including the names and sequences, in this study is provided in Supplementary Table 1.

**Flow cytometry**. The quantitative fluorescence of each cell line was estimated using a flow cytometer (BD Accuri™ C6 Plus flow cytometer; BD Biosciences, USA) with BD Accuri™ C6 Plus software (BD Biosciences, USA) and Flowjo software (BD Biosciences, USA). The cells were excited with a 488 nm laser and the emission was detected using a 533/30 nm bandpass filter (FITC-A). Details are provided in Supplementary Method 5. The gating strategy used for the analysis is described in Supplementary Fig. 9a–d.

**Cell tolerance test**. The pH values of the acidic media used in the low pH tolerance tests were titrated with HCl (both MTAP and MTP media). For the spot plating culture, liquid media at different pH values (i.e. pH 7.0 and pH 5.5) were solidified using plant agar (1.5% w/v) after autoclaving. Starting from $5 \times 10^5$ cells mL$^{-1}$, 4 serial tenfold dilutions were performed and 20 μL aliquots of each dilution were then spotted on the pH-adjusted plate. Moreover, a $CO_2$ tolerance test with an extremely high $CO_2$ condition (20%-enriched air) was conducted only with the autotrophic medium (i.e. MTP with an initial pH value of 7.0) to decouple the additional acidification effect, which may be caused by the organic carbon source (i.e. acetic acid)[38]. In addition, to exclude the adaptation effect, the cells were abruptly transferred from ambient conditions to high $CO_2$ conditions. The optical density ($OD_{800}$) of the cell suspension was converted into the biomass concentration (g L$^{-1}$) using the relevant calibration functions plotted for each strain (Supplementary Fig. 12). Linear regression curves were derived based on the correlation between the cell culture dry cell weight (DCW) and turbidity. The DCW was measured via a filter-based gravimetric method using a Whatman GF/C glass microfiber filter (GE Healthcare, USA)[45]. The cell samples were harvested over the course of the autotrophic cultures (at atmospheric pH 7.0 condition) of each cell line, which were separately conducted to establish the correlation. Aliquots of 100 μL and 5 mL from each culture were sampled to measure the turbidity and DW, respectively. The biomass productivity (P; g L$^{-1}$ d$^{-1}$) was then estimated using the following Eq. (2):

$$P = (X_F - X_I)/(t_F - t_I) \qquad (2)$$

where $X_F$ (g L$^{-1}$) and $X_I$ (g L$^{-1}$) indicate the final and initial DCW measured at two specific time points, namely, $t_F$ and $t_I$, respectively. Given the carbon balance under autotrophic conditions, the $CO_2$ fixation rate ($R_{CO2}$; g $CO_2$ L$^{-1}$ d$^{-1}$) was evaluated using the following Eq. (3)[2]:

$$CO_2 \text{ fixation rate } (R_{CO2}) = \frac{C}{100} \times \left(\frac{M_{CO_2}}{M_C}\right) \times P \qquad (3)$$

where C, $M_{CO_2}$, and $M_C$ denote the carbon content (% w/w), a molecular weight of $CO_2$ (44.01 g mol$^{-1}$), and atomic mass of carbon (12.01 g mol$^{-1}$), respectively. In the lab-scale tolerance test, the carbon content of each cell line was assumed to be 50%[45]. To accurately estimate the practical carbon fixation ability of the algal cells in the proof-of-concept outdoor cultivation, the carbon content of the microalgal pellet was actually determined using a 5E-CHN2200 elemental analyzer (Changsha Kaiyuan Instruments, China).

**Outdoor cultivation with coal combustion flue gas injection**. The microalgal $CO_2$ conversion plant for cultivating the engineered mutant (PMA4ΔCter-V4) was constructed and operated under the permission of the Korean government (license number: LML 19-912). In the demonstration plant (Fig. 4a), a proof-of-concept outdoor cultivation was conducted with a 10 L (working volume) bubble column PBR and an autotrophic medium previously optimized for outdoor microalgal cultivation[46]. Prior to microalgal cell inoculation, the culture system, including the PBR and 10 L of industrial water, was disinfected with high alkalinity (~pH 12; by supplementing 10 mM KOH), and the flue gas stream emitted from the neighboring coal combustion boiler was continuously sparged into the water phase at an aeration rate of 0.1 vvm[70]. The exhaust gas consisted of 80.57% $N_2$, 13.18% $CO_2$, 7.46% $O_2$, 20.27 ppm $NO_X$ and 32.04 ppm $SO_X$[46]. Because of the interaction of

KOH and acidic components in the flue gas (e.g. $CO_2$, $NO_X$, and $SO_X$), gas injection resulted in the neutralization of the aqueous phase (~pH 7.11) while automatically forming a bicarbonate buffer system that is suitable for large-scale outdoor cultivation[46,70]. The culture medium was then prepared by adding 7 mM $NH_4Cl$, 0.83 mM $MgSO_4 \cdot 7H_2O$, 0.45 mM $CaCl_2 \cdot 2H_2O$, 1.65 mM $K_2HPO_4$, 1.05 mM $KH_2PO_4$, 0.134 mM $Na_2EDTA \cdot 2H_2O$, 0.136 mM $ZnSO_4 \cdot 7H_2O$, 0.184 mM $H_3BO_3$, 40 μM $MnCl_2 \cdot 4H_2O$, 32.9 μM $FeSO_4 \cdot 7H_2O$, 12.3 μM $CoCl_2 \cdot 6H_2O$, 10 μM $CuSO_4 \cdot 5H_2O$, and 4.44 μM $(NH_4)_6MoO_3$. Based on the autotrophic culture broth, cultivation was initiated by the inoculation of 50 mg $L^{-1}$ freshly and atmospherically grown algal seeds and was continued for two weeks (14 days). The pH of the algal culture was maintained at ~6.56 throughout the operation (Supplementary Table 4). The resulting biomass was harvested by a continuous flow centrifuge (at $21,000 \times g$) and lyophilized (at a pressure of 1 kPa and a temperature of −55 °C) for further analyses.

**Analytical methods (total lipid content, total FAME content, and calorific value).** Using the harvested biomass from the outdoor cultivation, the lipid productivity (mg lipid $L^{-1}$ $d^{-1}$), biodiesel productivity (mg FAME $L^{-1}$ $d^{-1}$), and calorific productivity (kJ $L^{-1}$ $d^{-1}$) for each strain were estimated using the following Eqs. (4)–(6):

$$Lipid\ productivity = Total\ lipid\ content \times P \qquad (4)$$

$$Biodiesel\ productivity = Total\ FAME\ content \times P \qquad (5)$$

$$Calorific\ productivity = LHV \times P \qquad (6)$$

where P and LHV denote the biomass productivity (mg biomass $L^{-1}$ $d^{-1}$) and lower heating value (also known as net calorific value), respectively[45]. To calculate the various metrics, the P (mg biomass $L^{-1}$ $d^{-1}$), total lipid content (% w/w), total FAME content (% w/w), and LHV (kJ $g^{-1}$) values were determined. P was estimated using Eq. (2). To measure the total lipid content in the biomass, the total lipid fraction in the dried biomass was extracted according to a modified Bligh-Dyer method and the content was determined using the gravimetric assay. For the biodiesel content analysis, FAMEs (commonly referred to as biodiesel fractions) were prepared from the remaining total lipid fraction and analyzed using an Agilent 7890 A gas chromatography (GC) system (Agilent Technology, USA) equipped with a DB-23 GC column and flame ionization detector (FID). The extracted lipid sample was spiked with 10 mg of pentadecanoic acid (C15:0) as an internal standard and then converted into FAMEs through transesterification using 3% methanolic sulfuric acid. The resulting converted lipids were then quantified by GC. The Supelco® 37 Component FAME Mix, F.A.M.E. Mix RM-3, and F.A.M.E. Mix RM-5 (all of which were purchased from Sigma-Aldrich, USA) were adopted as the FAME reference standard. Helium and a mixture of hydrogen and high purity air were used as the carrier gas and fuel gas for FID, respectively. The GC operation conditions were as follows: injected volume, 1 μL; split ratio, 1:50; inlet temperature, 250 °C; detector temperature, 280 °C; and oven temperature, maintained at 50 °C for 1 min, increased to 175 °C at a rate of 25 °C $min^{-1}$, increased to 230 °C at a rate of 4 °C $min^{-1}$, and maintained at 230 °C for 5 $min^{32}$. The evaluation of calorific value (in LHV) of the biomass was performed using a 5E-C5500 automatic bomb calorimeter (Changsha Kaiyuan Instruments, China).

**Statistics and reproducibility.** The replicates of all experiments were indicated in the corresponding figure legends. The results are represented as the mean values and error bars, which show the standard deviations of the mean (mean ± SD). Where necessary, a two-tailed Student's *t*-test was conducted to determine whether statistically significant differences existed between the mean values of the two groups. In the analysis of RNA-seq data, the genes with $|log_2FC| > 1$ and $p < 0.05$ were considered DEGs. The statistically significant differences were determined by the generalized linear model (GLM) likelihood ratio (LR) test. Overrepresented GO terms were determined based on a cutoff value of $p < 0.05$ using Fisher's exact test method. The $pH_i$ values of the engineered strains and WT control strain were compared with box and whisker plots to demonstrate the median values, maximum and minimum points, and interquartile ranges. Gel and blot images, including PCR and immunoblotting results, were obtained from a single replicate. In quantitative analysis of protein expression levels, comparison among the protein samples is performed within the same gel. Confocal images were acquired from three biologically independent samples.

**Reporting summary.** Further information on research design is available in the Nature Research Reporting Summary linked to this article.

## Data availability

Data supporting the findings of this work are available within the paper and its Supplementary Information files. A reporting summary for this article is available as a Supplementary Information file. All sequence reads for the RNA-seq were deposited in the NCBI Sequence Read Archive (SRA) with BioProject accession code PRJNA720171. The updated *Chlamydomonas reinhardtii* genome and gene annotation for the RNA sequencing and DNA insertion analysis were downloaded from JGI Phytozome v13,

*Chlamydomonas reinhardtii* v5.6 database [https://data.jgi.doe.gov/refine-download/phytozome?organism=Creinhardtii&expanded=281]. All sequence reads for the insertion site were deposited in the NCBI SRA with BioProject accession code PRJNA767110. Source data are provided with this paper.

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

## Acknowledgements

We are grateful for the funds provided by Korea Carbon to X R&D Center (2020M3H7A1098295) and the National Research Foundation (NRF-2019R1A2C3009821 / NRF-2020R1A5A1018052) of the Ministry of Science and ICT of Korea. The authors express special thanks to Korea University and Korea Western Power Co., Ltd. for supporting this research. We are also appreciative of Hong Ki Yoon at Korea University, Hyunjin Ko at the Korea Western Power Co., Ltd., and Dr. Jong-Kyun You and Dr. Dae Ho Lim at the Korea Institute of Energy Research for helping with the outdoor microalgal cultivation.

## Author contributions

S.J.S. supervised the whole project and reviewed the entire manuscript. H.I.C. and S.-W.H. designed and performed the experiments and wrote the manuscript. J.K. and E.S. conducted the transgene design and protein expression verification. B.P. and I.-G.C. carried out the RNA sequencing, relevant statistical processing, and insertion site analysis. All the authors provided remarks on the work and checked and proofread the final version of the paper.

## Competing interests

The authors declare no competing interests.
