## [Peer Review File · Nature Communications]

Augmented CO₂ tolerance by expressing a single H⁺-pump enables microalgal valorization of industrial flue gasREVIEWER COMMENTS

Reviewer #1 (Remarks to the Author):

The paper under review reflects, in my opinion, the first conclusive evidence for the crucial role of pH in the development of high CO₂-induced damage in microalgae. On the other hand, the authors have shown unequivocally that improving the proton-extrusion capability of the cell would be sufficient for the development of high CO₂-tolerance. Noteworthy, these mechanisms were long discussed for recent three decades, but it seems that only in this study the conclusive proof obtained in systemic manner has been presented. The experimental design is comprehensive; the interpretation of the results is sound and complete leaving no opening for doubt. The text is well-written, some minor flaws in style and grammar could be corrected at copyediting stage. The only potential deficiency of the paper I can highlight is related with incompatibility of the approach suggested (engineering of the strains) with bulk biomass production. Specifically, the release of the engineered organisms into environment is still legally prohibited. In effect, this confines the strains with fortified pH tolerance to closed cultivation systems (PBRs). This approach is economically viable for obtaining high-valued products but for bulk products e.g. biomass for the conversion to biofuel might not be so. The authors are encouraged to discuss this obstacle.

Reviewer #2 (Remarks to the Author):

The manuscript by Choi et al. describes micro algae-based CO₂ conversion technology by genetic engineering. The authors found that the down-regulation of PMAs and the resulting intracellular acidification is a major toxicity mechanism under high CO₂ environments. Therefore, the authors constructed microalgal strains expressed exogenous PMA, and confirmed whether the strains grew well under acidic or High CO₂ condition. In addition, the authors conducted POC cultivation and found that the productivity of the transformant strain was better than that of the wild strain. Overall this study sounds interesting and the scope is understandable.

My comments are below:

1. Transcriptome analysis is based on duplication, not triplication.
It is generally accepted that a minimum of three replicates is required to perform a statistically reliable transcriptome analysis.
Therefore, we do not know whether this transcriptome analysis is statistically meaningful or not.
2. It is important that all data be easily accessible and searchable so that the transcriptome data can be used by other groups.
3. The authors should describe what kind of error bars (SD? SE?) on the growth curve and bar graphs in the figures, and how many experiments it is based on.
4. The authors confirmed that the expression of PMA2 and PMA3 genes was decreased under low pH and high CO₂ conditions by RNAseq analysis.
In SFig.3b, the authors have confirmed that the expression of PMA3 gene was down-regulated under high CO₂ conditions by qRT-PCR.
I think that the expression level of both PMA genes under low pH and high CO₂ conditions should also be confirmed by qRT-PCR.
5. Are there any distinctive differences between the sequence and function of PMA2 and PMA3 genes?
6. The PMA4ΔCter-V, which is C-terminal truncated mutant (probably highly active), was used in this

study.

Do the PMAs of *C. reinhardtii* also have autoinhibitory domain in the C-terminus?

Do you have any knowledge about the regulation of PMA activity under various conditions (e.g. low pH)?

You might want to explain this.

7. The authors detected exogenous expressed PMA4 Δ Cter-V using by anti H⁺-ATPase antibody (AS07260).

The anti-H⁺-ATPase antibody has been confirmed to react with H⁺-ATPase of *C. reinhardtii* according to information published by Agredera.

I wonder why you have not compared the protein level of PMA4 Δ Cter-V with that of the native H⁺ATPase.

That is important information to understand the characteristics of these transformants.

8. The authors showed that the down-regulation of PMAs under low pH or high CO₂ condition by transcriptome analysis (Fig. 1b).

You should confirm the differences in protein levels of H⁺-ATPase between low pH and neutral conditions (high CO₂ and ambient air conditions).

9. Did you measure the pH values in POC open cultivation?

I understand how difficult to test the large-scale cultivation repeatedly.

If you have the data, you might want to present that.

10. In Supplementary Table6, CO₂ fixation rate (g CO₂ L⁻¹ d⁻¹) > (mg CO₂ L⁻¹ d⁻¹) ?

11. This reviewer thinks that authors should avoid "data not shown" (Page 23, line 563, Supplementary Information, Page 3 line 89).

Reviewer #3 (Remarks to the Author):

This manuscript investigates the toxic effects of extremely high (20%) CO₂ on *Chlamydomonas* cells. Plasma membrane H⁺-ATPase (PMA) was identified as a target for strain engineering, and transgenic lines expressing a heterologous PMA were made that are able *Chlamydomonas* to tolerate low pH and grow photoautotrophically in 20% CO₂. Remarkably, these lines exhibited improved photoautotrophic growth when provided with flue gas from a coal-fired power plant. This is an interesting finding that opens the door to engineering of other, more industrially relevant microalgae. However, there are some issues with replication and statistical analysis that need to be resolved, as described below.

1. Fig. 1a: This figure shows growth data for only two time points using OD800, and it is not clear if these points represent the exponential growth phase. How could this analysis be used to select "conditions for EC50" as stated in the legend? A much more thorough analysis of growth is necessary, with cell counts and/or ash-free dry weight measurements, at least three biological replicates, and a full growth curve with calculation of exponential growth rates.

2. Fig. 1b is cited in support of the statement that high CO₂ and low pH do not share the same toxicity mechanisms (lines 93-95). It is not clear how this is evident from the figure. Fig. 1c shows overlap between the genes expressed in high CO₂ and low pH.

3. The RNA-seq analysis that led to identification of PMA as a target has a major flaw in its experimental design. According to the Methods, the RNA-seq analysis was done with a single sample (consisting of a pool of two biological replicates) from each condition. This makes any meaningful statistical analysis of differentially expressed genes impossible, and I cannot understand how p values could be calculated. One way to resolve this issue would be to repeat the RNA-seq experiment with at

least three biological replicates and a proper statistical analysis. Another way would be to treat the RNA-seq experiment as only a screening approach to identify possible candidates. In this case, the Results text would have to be heavily edited, and no conclusions should be drawn from the RNA-seq data about differential gene expression, specific groups of genes (i.e., CCM genes), or GO enrichment. Any specific genes of interest that are mentioned in the text would need to have their expression quantified by qPCR. In the current manuscript, only one gene (PMA3) was analyzed by qPCR (Suppl. Fig. 3).

4. lines 149-154: It is unlikely that downregulation of CCM genes is "due to reducing further endomembrane acidification," unless this were the signal for regulation of CCM genes, for which there is no evidence that I am aware of. Most CCM genes are expressed in low (ambient) CO₂ but not high CO₂, because they are controlled by the regulator CCM1/CIA5.

5. Figure legends should specify the number of biological replicates for each experiment. Preferably, this should be at least three in each experiment.

6. How does the photoautotrophic productivity of the engineered lines (and wild type) in extremely high CO₂ (20% CO₂ or flue gas) compare with productivity in ambient CO₂?

7. Data availability: All RNA-seq reads should be made available through a publicly accessible repository (e.g. NCBI GEO), and an accession number should be provided.

8. Overall, the manuscript needs extensive copyediting for English grammar and readability. I have noted a few instances in the minor comments below, but there are many more.

Minor comments:

9. Abstract, line 22: "prone to CO₂" does not make sense

10. Abstract, line 26: "first time demonstrated" sounds awkward

11. Abstract, line 30: "practical toxic" does not make sense

12. line 45: should specify "high CO₂"

13. line 69: do you mean "too low" instead of "low enough"?

14. line 81: delete "lucrative"

15. Fig. 1c: Can this figure be replotted so that the areas in the Venn diagrams are proportional to the number of genes?

16. line 195: specify that PMA3 is "the gene of interest"

17. line 258: delete "vivid"

18. Suppl. Fig. 5: legend should state that WT cells are shown in panel (b) instead of panel (a)

19. line 415: change "practical" to "actual"

Point-by-Point Response to the Reviewers' Comments of

Manuscript NCOMMS-21-11380A

Title: Augmented CO₂ tolerance by expressing a single H⁺-pump enables microalgal valorization of industrial flue gas

First of all, we would like to express our thanks for your time and efforts in reviewing our manuscript. Based on your valuable and detailed comments, we have addressed and responded all issues indicated in the referees' comment. In addition, we have voluntarily tried to find and revise all the typos throughout the paper. Taken together, herein, we provide a point-by-point response to your comments. The revised parts are indicated in red in the main manuscript. In this response letter, for your readability, our response to each of your comments are presented in *blue-colored italic letters* following your comments. Once Again, we are very appreciative of your great interest in our manuscript and sincerely hope you are satisfied with our revised manuscript. Your comments and questions are invaluable and undoubtedly of help not only to this article but also to our present and future research works.

Main changes in the Revision Process

- 1. Discussions on the potential problems regarding outdoor cultivation of genetically modified microalgae, including biocontainment and economic aspects, were added.*
- 2. The number of replicates in the RNA-Seq-based transcriptome analysis was increased from $n = 2$ to $n = 6$ for confirmation of the reproducibility and improvement of the statistical meaningfulness.*
- 3. The immunoblotting (Western blotting) assay was newly performed for verification of condition-dependent expression the *Chlamydomonas reinhardtii* native PMA at the protein levels.*
- 4. The number of replicates of each experiment is clearly presented in the corresponding figure legends and the "Statistical analysis" section in the "Methods." Several display items were repotted as dot plots to clearly show the distribution of the underlying data (the data are provided in the "Source Data" file).*
- 5. Newly added Supplementary Figures have been cited in the manuscript and all references to the figures were reordered during the revision.*
- 6. While thoroughly inspecting the raw data during the course of preparing the Source Data file, the data in the study were double-checked and rectified.*
- 7. The manuscript and the Supplementary Information were thoroughly copyedited for correcting typos and improving English grammar and readability.*

Comments from Reviewer #1

The paper under review reflects, in my opinion, the first conclusive evidence for the crucial role of pH in the development of high CO₂-induced damage in microalgae. On the other hand, the authors have shown unequivocally that improving the proton-extrusion capability of the cell would be sufficient for the development of high CO₂-tolerance. Noteworthy, these mechanisms were long discussed for recent three decades, but it seems that only in this study the conclusive proof obtained in systemic manner has been presented. The experimental design is comprehensive; the interpretation of the results is sound and complete leaving no opening for doubt. The text is well-written, some minor flaws in style and grammar could be corrected at copyediting stage. The only potential deficiency of the paper I can highlight is related with incompatibility of the approach suggested (engineering of the strains) with bulk biomass production. Specifically, the release of the engineered organisms into environment is still legally prohibited. In effect, this confines the strains with fortified pH tolerance to closed cultivation systems (PBRs). This approach is economically viable for obtaining high-valued products but for bulk products *e.g.*, biomass for the conversion to biofuel might not be so. The authors are encouraged to discuss this obstacle.

- *We appreciate your valuable comment. As you concerned, the need to use PBRs for biocontainment to prevent undesired dissemination of genetically modified (GM) microalgae to the environment and the resulting increase in production costs can be a negative factor which deteriorates the product's economic competitiveness. Although the cultivation of GM algae in open environment does not seem to be accompanied by greater environmental risks than expected according to the previous studies (Szyjka et al., Algal Res., 2017), it is still legally prohibited and public concerns on undesired dissemination of GM organisms are not negligible. Therefore, PBR systems should be inevitably exploited for the cultivation of GM algae. In fact,*

*according to an extensive techno-economic analysis study of microalgal cultivation, PBR can be used as equipment for moderating the product's price in the near future with ongoing technical developments (Ruiz et al., Energy Environ. Sci., 2016). Their use obviously ensures better photosynthetic efficiency and shorter light paths that are key parameters lowering the biomass production cost. These properties eventually contribute to improving the biological process' economic feasibility by counterweighting the cost for the PBR installation and operation. Besides, considering the goal of the cultivation – practical CO₂ reduction with phototrophic microalgal culture, PBR constitute the most suitable platform for serving the purpose of enabling intensive bioconversion of CO₂. Thus, harnessing well-designed and cost-effective PBRs may simultaneously guarantee thorough containment of GM organisms, efficient CO₂ fixation, and high biomass productivity while securing economic competitiveness. Please refer to the revised manuscript **lines 567–584**, where the related discussion is reflected.*

- Plus, we have thoroughly copyedited the sentences throughout the manuscript and the Supplementary Information for improved English grammar and readability. Please find the copyedited parts which is highlighted with red color.*

References

- 1. Szyjka, S. J., et al. Evaluation of phenotype stability and ecological risk of a genetically engineered alga in open pond production. *Algal Res.* **24**, 378–386 (2017).*
- 2. Ruiz, J., et al. Towards industrial products from microalgae. *Energy Environ. Sci.* **9**, 3036–3043 (2016).*

Comments from Reviewer #2

The manuscript by Choi *et al.* describes micro algae-based CO₂ conversion technology by genetic engineering. The authors found that the down-regulation of PMAs and the resulting intracellular acidification is a major toxicity mechanism under high CO₂ environments. Therefore, the authors constructed microalgal strains expressed exogenous PMA, and confirmed whether the strains grew well under acidic or high CO₂ condition. In addition, the authors conducted POC cultivation and found that the productivity of the transformant strain was better than that of the wild strain. Overall, this study sounds interesting and the scope is understandable. My comments are below:

1. Transcriptome analysis is based on duplication, not triplication. It is generally accepted that a minimum of three replicates is required to perform a statistically reliable transcriptome analysis. Therefore, we do not know whether this transcriptome analysis is statistically meaningful or not.

- *We are grateful for your constructive comment. Even though there are several microalgal RNA-Seq duplication examples (Jaeger et al., Biotechnol. Biofuels, 2017, Zienkiewicz et al., Plant Physiol., 2020), which we previously referred to, we entirely agree with your concern. To dispel your and potential reader's worries on the statistical reliability, we newly conducted 4 more replicates of the transcriptome analysis (total n = 6). Consequently, we found that the results of the GO enrichment analysis and investigation on the individual gene expression were highly retained (Fig. 1d, Supplementary Figs. 2–3, and Supplementary Table 1) compared to the conclusive data obtained from the previous analysis with n = 2, and successfully reached the same conclusion that downregulation of PMA genes under high CO₂ conditions would be the primary CO₂ toxicity mechanism of the CO₂-intolerant green microalga, *C. reinhardtii*, with verified reproducibility and improved statistical*

meaningfulness. Please find the detailed discussions in the revised manuscript lines 94–95, 669–670, and 829–838.

2. It is important that all data be easily accessible and searchable so that the transcriptome data can be used by other groups.

- *We thank for your comments. As you requested, the transcriptome data in this study were deposited on the public repository (the NCBI Sequence Read Archive; SRA) to make them easily accessible and searchable. Please find the dataset with the accession numbers of SRR14160260, SRR14160261, SRR15243031, SRR15243032, SRR15243035, and SRR15243036 (six replicates of ambient conditions-grown cells; atmospheric pH 7.0), SRR14160258, SRR14160259, SRR15243027, SRR15243028, SRR15243029, and SRR15243030 (six replicates of low pH conditions-grown cells; pH 5.0), and SRR14160256, SRR14160257, SRR15243025, SRR15243026, SRR15243033, and SRR15243034 (six replicates of high CO₂ conditions-grown cells; 20% CO₂-enriched air).*

3. The authors should describe what kind of error bars (SD? SE?) on the growth curve and bar graphs in the figures, and how many experiments it is based on.

- *Thank you for your comment. All data was displayed with their standard deviations (SDs). According to your recommendation, we have clarified the kind of error bars and the number of experimental replicates in the individual figure legends (both main and supplementary figures) and in the "Statistical analysis" in the "Methods" part. Please refer to the revised manuscript lines 818–826, 1046–1119, and Supplementary Information lines 283–395, where the related revisions are reflected.*

4. The authors confirmed that the expression of PMA2 and PMA3 genes was decreased under low pH and high CO₂ conditions by RNAseq analysis. In SFig.3b, the authors have confirmed that the expression of PMA3 gene was down-regulated under high CO₂ conditions by qRT-PCR. I think that the expression level of both PMA genes under low pH and high CO₂ conditions should also be confirmed by qRT-PCR.

- *The authors appreciate your comment. We previously verified only PMA3 gene's mRNA expression level by qRT-PCR because that was the sole gene which was significantly downregulated under acidic milieus in which we are interested. However, for better reliability of the study as you recommended, we have observed the expression of both PMA genes, namely PMA2 and PMA3, at the transcriptome level under low pH and high CO₂ conditions using qRT-PCR. Resultingly, as shown in **Supplementary Figs. 6a–6b**, we found that the expression levels of the genes which were individually obtained from both measurements showed the same gene regulation trends, thereby successfully confirming the reproducibility of the expression levels of the genes. Please find the related discussion in the revised manuscript **lines 213–218**.*

5. Are there any distinctive differences between the sequence and function of PMA2 and PMA3 genes?

- *Thank you for your inspirational question. Since there is no exhaustive study on the discrimination of the difference between two plasma membrane H⁺-ATPases (PMAs) so far, we conducted a multiple sequence alignment analysis with the protein sequences. The newly shown alignment test (**Supplementary Fig. 7**) with *C. reinhardtii* PMA2 (Cre10.g459200) and PMA3 (Cre03.g164600)*

compared with model plasma membrane H^+ -ATPases – AHA2 and PMA4 which are an autoinhibited plasma membrane H^+ -ATPase in *Arabidopsis thaliana* and a plasma membrane H^+ -ATPase in *Nicotiana plumbaginifolia*, respectively – revealed that the algal proteins commonly have conserved Asn106 (N106) and Asp684 (D684) analogy (sequential numbering based on the AHA2 sequence) which are known to play a pivotal role in H^+ transport (Palmgren, *Annu. Rev. Plant Physiol. Plant Mol. Biol.* 2001; Pedersen et al., *Nature*, 2007). In addition, both algal proteins (i.e., PMA2 and PMA3) are also found to contain a highly conserved region (DKTGTLT) that is one of the signature characteristics of P-type ATPase (Palmgren, *Annu. Rev. Plant Physiol. Plant Mol. Biol.* 2001), all of which clearly shows the PMAs serve as H^+ -pumps. Previous reports also show that the proteins are obviously H^+ -pumps located on the plasma membrane (Campbell et al., *J. Phycol.*, 2001; Hirooka et al., *Proc. Natl. Acad. Sci. U.S.A.*, 2017; Mackinder et al., *Cell*, 2017; Norling et al., *Physiol. Plant*, 1996).

· Our experimental data on the gene expression levels – PMA2 showed opposite regulation patterns against CO_2 -dependent (downregulated) and independent (upregulated) conditions whereas PMA3 was always downregulated under both acidic stress conditions – implies that there is a distinguishing function between two PMAs especially under CO_2 -independent acidic conditions. We can predict the sequence variation (**Supplementary Fig. 7**; identity of 31.9% and similarity of 47.4%) would be a factor which may confer a distinctive role of each PMA, but because the details on the distinctive roles of the pumps have not been clearly explained as mentioned, further studies are required to accurately discriminate the respective roles of PMA2 and PMA3.

- *On the one hand, interestingly, PMA2 and PMA3 seem intimately related to the carbon concentrating mechanism (CCM) of the alga, C. reinhardtii, according to Mackinder et al. Especially, PMA2 was revealed from the study to form a complex with LCII and HLA3 (bicarbonate transporters), both of which are core elements of the CCM. We thus speculate that the relevance of the PMAs with the CCM could be of help to explain why the PMAs are downregulated under high CO₂ conditions. Please refer to the revised manuscript **lines 218–221, 517–531, and Supplementary Fig. 7** where the related discussions are reflected.*

6. The PMA4ΔCter-V, which is C-terminal truncated mutant (probably highly active), was used in this study. Do the PMAs of *C. reinhardtii* also have autoinhibitory domain in the C-terminus? Do you have any knowledge about the regulation of PMA activity under various conditions (e.g., low pH)? You might want to explain this.

- *We appreciate your motivational question. As stated in the manuscript, a model plant PMA, PMA4, includes the auto-inhibitory domain in the C-terminus. The autoinhibitory domain is known to be regulated by 14-3-3 proteins. The domain is tipped with a signature sequence, His/Ser-Tyr-pThr(phosphorylated Thr)-Val-COOH (penultimate threonine-containing sequence; pT), where the regulatory 14-3-3 protein binds. It is also known that the peptide sequences, Regions I and II (R-I and R-II) also significantly influence on the autoinhibitory functionality (Okumura et al., Plant Physiol., 2012; Zhang et al., Front. Plant Sci., 2020). According to our alignment analysis (**Supplementary Fig. 7**), the unique peptide sequences (i.e., pT, R-I, and R-II) are highly conserved in two model terrestrial plant-originated PMAs whereas they are*

*not conserved in the ancestral algal PMAs (i.e., PMA2 and PMA3). A previous studies on evolutionary analysis of PMAs suggests the sequence may be formed in the course of their revolutionary processes (Zhang et al., Front. Plant Sci., 2020). Even though the microalgal PMAs do not include such noticeable characteristic sequences, there is a high possibility that C. reinhardtii PMAs' C-terminus may also serve an important role in regulating the activity considering previous reports (Lecchi et al., J. Biol. Chem., 2007; Okumura et al., Plant Physiol., 2012). Still, the exact regulatory mechanism is not revealed yet. In order to clarify this, further comprehensive biochemical experiments should be carried out in parallel with the research for identifying a distinctive function between two native PMAs (PMA2 and PMA3; please refer to the response 5) of the microalga. Thus, at this moment, we just can anticipate that the H⁺ pumping ability of C. reinhardtii may mainly be adjusted by regulating RNA and protein expression levels of the related genes in consideration of the RNA-Seq data and immunoblotting results in this study (Fig. 1d and Supplementary Fig. 6; please also refer to the response to your comment 8). Please find the related discussions in the revised manuscript **line 249–254, 531–536, and Supplementary Fig. 7.***

7. The authors detected exogenous expressed PMA4ΔCter-V using by anti H⁺-ATPase antibody (AS07 260). The anti-H⁺-ATPase antibody has been confirmed to react with H⁺-ATPase of *C. reinhardtii* according to information published by Agrisera. I wonder why you have not compared the protein level of PMA4ΔCter-V with that of the native H⁺-ATPase. That is important information to understand the characteristics of these transformants.

We deeply thank for your constructive comment. Previously, we detected the exogenously expressed with a total protein of 20 μg extracted from each cell

line, which is a generally adopted amount used in typical Western blot analysis as stated in technical protocols published by Agrisera (please refer to the detailed process in **Supplementary Methods**). Consequently, as shown in **Fig. 2e**, a band at approximately 130 kDa (the expected molecular weight of the exogenous protein: 120.81 kDa) was visualized only with the transgenic strains indicating that the overexpression of the PMA successfully occurred. The result also implies that the expression of the native PMA is too low to be detected by the method.

- Considering this, for verification of the PMA downregulation at the protein levels under stressful conditions in the WT strain (i.e., to fulfill the suggestion casted from your comment 8), we then loaded 10 times more proteins (i.e., 200 μ g) to detect the native PMA and quantitatively compare the exposure condition-dependent PMA expression levels. As a result, we could detect the endogenous PMA and compare the expression levels depending on the cell culture conditions (**Supplementary Fig. 6c and the attached figure below**). Please find the related descriptions and discussions in the revised manuscript **lines 213–218 and 287–292, and Supplementary Information lines 103–147 (in the Supplementary Methods) and 323–335 (in the figure legend of Supplementary Fig. 6)**.
- Regarding the comparison of the protein expression levels of the exogenous PMA with the native PMA, unfortunately, it is hardly possible to separately quantify the respective protein expressions using the immunoblotting method. This is because the molecular weights of the exogenously expressed PMA (ca. 120.81 kDa) and the native PMA (ca. 117.89 kDa) are highly similar, which inevitably hinders their separation and quantitation.

Immunoblotting analysis for the detection of the exogenously expressed PMA and the verification of the native PMA expression at the protein levels. Considering the low abundance of the native PMA in the WT strain, the protein loading amount was increased. MW_e represents the expected molecular weight of the corresponding proteins which can be estimated from their amino acid sequences. The uncropped images are provided in Source Data file.

8. The authors showed that the down-regulation of PMAs under low pH or high CO₂ condition by transcriptome analysis (Fig. 1b). You should confirm the differences in protein levels of H⁺-ATPase between low pH and neutral conditions (high CO₂ and ambient air conditions).

- We are much appreciated for your constructive comment. We totally agree with your point that emphasizes the importance of confirming the protein expression level. In this regard, we have newly performed immunoblotting (Western blot) analysis to confirm the differences in the protein expression levels of H⁺-ATPase depending on the external conditions using the anti H⁺-ATPase antibody (AS07 260, Agrisera). In order to detect and compare the band intensity, we loaded a much higher total protein amount (i.e., 200 μ g) compared with the previous detection considering the low abundance of the native PMA (please refer to **the attached figure** beneath the response to your comment 7). We then successfully verified that the native H⁺-ATPase is also downregulated at the protein levels which corresponds to the downregulation of RNA expression. From this result, we could verify the mRNA-protein expression correspondence and infer that the downregulation can directly influence the congenital intolerance. Please find the results and discussion in the revised manuscript lines 213–218 and Supplementary Fig. 6c.*

9. Did you measure the pH values in POC open cultivation? I understand how difficult to test the large-scale cultivation repeatedly. If you have the data, you might want to present that.

- *First of all, thank you for your understanding. Fortunately, we have the data on the initial (measured immediately before the seeding) and final pH (measured right before the harvesting) values of the POC microalgal outdoor cultivations. Please find the related contents in the revised manuscript **lines 775 and 783, and Supplementary Table 6.***

10. In Supplementary Table 6, CO₂ fixation rate (g CO₂ L⁻¹ d⁻¹) > (mg CO₂ L⁻¹ d⁻¹)?

- *Thank you for your detailed comment. We have corrected the unit according to your point. At the same time, the authors have also voluntarily revised the unit of calorific productivity (from kJ g⁻¹ d⁻¹ to kJ L⁻¹ d⁻¹) in the same table during our thorough manuscript inspection which was miss-typed before. Please find the corrections in **Supplementary Table 6.***

11. This reviewer thinks that authors should avoid "data not shown" (Page 23, line 563, Supplementary Information, Page 3 line 89).

- *We deeply appreciate your recommendation. According to your recommendation that "data not shown" should be avoided, we have displayed all related data and images that had been presented as "data not shown" in the previously submitted version.*

- ✓ Page 23 line 563 (data not shown) → Shown in **Supplementary Fig. 1**: Condition screening experimental data for EC_{50} determination.
- ✓ Supplementary Information page 3 line 89 (data not shown) → Shown in **Supplementary Figs. 8b–8c**: Confirmation of the gene insertion site of the transgenic strains (PMA4ΔCter-V4 and PMA4ΔCter-V10) using PCR.

References

1. Jaeger, D. et al. Time-resolved transcriptome analysis and lipid pathway reconstruction of the oleaginous green microalga *Monoraphidium neglectum* reveal a model for triacylglycerol and lipid hyperaccumulation. *Biotechnol. Biofuels* **10**, 197 (2017)
2. Zienkiewicz, A. et al. The microalgal *Nannochloropsis* during transition from quiescence to autotrophy in response to nitrogen availability. *Plant Physiol.* **182**, 819–839 (2020)
3. Palmgren, M. G. Plant plasma membrane H^+ -ATPases: Powerhouses for nutrient uptake. *Annu. Rev. Plant Physiol. Plant Mol. Biol.* **52**, 817–845 (2001).
4. Pedersen, B. P., Buch-Pedersen, M. J., Morth, J. P., Palmgren, M. G., Nissen, P. Crystal structure of the plasma membrane proton pump. *Nature* **450**, 1111–1119 (2007).
5. Campbell, A. M., et al. Identification and DNA sequence of a new H^+ -ATPase in the unicellular green alga *Chlamydomonas reinhardtii* (Chlorophyceae). *J. Phycol.* **37**, 536–542 (2001).
6. Hirooka, S. et al. Acidophilic green algal genome provides insights into adaptation to an acidic environment. *Proc. Natl. Acad. Sci. U.S.A.* **114**, 8304–8313 (2017).

7. Mackinder, L. C. M., et al. *A Spatial Interactome Reveals the Protein Organization of the Algal CO₂-Concentrating Mechanism. Cell* **171**, 133–147 (2017).
8. Norling, B., Nurani, G., Franzen, L. G. *Characterisation of the H⁺-ATPase in plasma membranes isolated from the green alga Chlamydomonas reinhardtii. Physiol. Plant.* **97**, 445–453 (1996).
9. Okumura, M., Inoue, S., Takahashi, K., Ishizaki, K., Kohchi, T., Kinoshita, T. *Characterization of the Plasma Membrane H⁺-ATPase in the Liverwort Marchantia polymorpha. Plant Physiol.* **159**, 826–834 (2012).
10. Zhang et al. *Evolutionary and functional analysis of a Chara plasma membrane H⁺-ATPase. Front. Plant Sci.* **10**, 1707 (2020)
11. Lecchi et al. *Tandem phosphorylation of Ser-911 and Thr-912 at the C terminus of yeast plasma membrane H⁺-ATPase leads to glucose-dependent activation. J. Biol. Chem.* **282**, 35471–35481 (2007).

Comments from Reviewer #3

This manuscript investigates the toxic effects of extremely high (20%) CO₂ on *Chlamydomonas* cells. Plasma membrane H⁺-ATPase (PMA) was identified as a target for strain engineering, and transgenic lines expressing a heterologous PMA were made that are able *Chlamydomonas* to tolerate low pH and grow photoautotrophically in 20% CO₂. Remarkably, these lines exhibited improved photoautotrophic growth when provided with flue gas from a coal-fired power plant. This is an interesting finding that opens the door to engineering of other, more industrially relevant microalgae. However, there are some issues with replication and statistical analysis that need to be resolved, as described below.

1. Fig. 1a: This figure shows growth data for only two time points using OD₈₀₀, and it is not clear if these points represent the exponential growth phase. How could this analysis be used to select “conditions for EC₅₀” as stated in the legend? A much more thorough analysis of growth is necessary, with cell counts and/or ash-free dry weight measurements, at least three biological replicates, and a full growth curve with calculation of exponential growth rates.

- *The authors appreciate your accurate point. We have defined EC₅₀ values in this study as effective concentrations that causes decreases in the initial growth by 50% (compared with that under ambient conditions; previously estimated from optical density change, namely ΔOD₈₀₀, as a measure of biomass increment) during a short period of time (for a day). This was chosen to reveal the cellular response against the stressful stimuli (i.e., low pH and high CO₂) before the algal cells completely enter their adaptive stages (Kawasaki et al., Plant Cell, 2001). A faster growth rate under 20% CO₂ condition compared with that under ambient condition (between Day 3 and 4;*

Supplementary Fig. 1) implies there exist high CO₂ stress-related adaptive responses that temporarily enable the algal cells to utilize abundant inorganic carbon sources in the surroundings efficiently. In the end, however, the cells came to death within six days of cultivation due to a failure to get over the continuously exerted stress.

*To clearly show the EC₅₀ determining process, we have thoroughly re-measured the microalgal growth data in terms of the OD₈₀₀ values, cell numbers, and dry cell weights, according to your comment (only optical densities were previously measured). Then, full growth curves based on the turbidities, cell counts, and dry weights (with three biological replicates) under various conditions could be compiled in **Supplementary Fig. 1**. As a result, like before, we could reconfirm that pH 5.0 and CO₂ 20% are right conditions which trigger 50% cell growth (i.e., biomass increment) inhibitions. Furthermore, as shown in the figure, the optical densities clearly reflected the biomass concentrations of the culture (i.e., dry cell weight) during the period of EC₅₀ observation (**Supplementary Figs. 1b, 1d, and 1f**), which suggests that the turbidity (OD₈₀₀) can be alternatively and conveniently measured to investigate the algal culture's biomass concentration (an actual indicator of the biomass increment) on behalf of direct estimation of dry cell weight (by filter-mediated gravimetric analysis), at least in this analysis case. Please refer to the revised manuscript **lines 91–94 and 655–665**, where the related discussion is reflected.*

2. Fig. 1b is cited in support of the statement that high CO₂ and low pH do not share the same toxicity mechanisms (lines 93-95). It is not clear how this is evident from the figure. Fig. 1c shows overlap between the genes expressed in high CO₂ and low pH.

- *We are appreciative of your insightful comment. First of all, in order to clarify the meaning of the statement, we have rephrased the sentence of the originally submitted manuscript line 95. In addition, while having replotted **Fig. 1c** as you also suggested (your comment 15), we have added several discussions to support our speculation that high CO₂ and low pH conditions do not seem to share the exact same toxicity mechanisms. Please refer to the revised manuscript **lines 95–100 and 135–139**, where the related discussion is reflected.*

3. The RNA-seq analysis that led to identification of PMA as a target has a major flaw in its experimental design. According to the Methods, the RNA-seq analysis was done with a single sample (consisting of a pool of two biological replicates) from each condition. This makes any meaningful statistical analysis of differentially expressed genes impossible, and I cannot understand how *p* values could be calculated. One way to resolve this issue would be to repeat the RNA-seq experiment with at least three biological replicates and a proper statistical analysis. Another way would be to treat the RNA-seq experiment as only a screening approach to identify possible candidates. In this case, the Results text would have to be heavily edited, and no conclusions should be drawn from the RNA-seq data about differential gene expression, specific groups of genes (*i.e.*, CCM genes), or GO enrichment. Any specific genes of interest that are mentioned in the text would need to have their expression quantified by qPCR. In the current manuscript, only one gene (PMA3) was analyzed by qPCR (Suppl. Fig. 3).

- *The authors are appreciated for your constructive comment. Even though there are several microalgal RNA-Seq analyses insisting their statistical significance*

with duplication samples (Jaeger et al., *Biotechnol. Biofuels*, 2017, Zienkiewicz et al., *Plant Physiol.*, 2020), we strongly agree that the small number of the experimental replications could raise serious concerns about the reliability of the transcriptome analysis result. In order to dispel the apprehension, we have newly conducted RNA-Seq with 4 more samples for improving the statistical meaningfulness. Namely, the statistical analysis with RNA-Seq data was totally redone with six biological triplicates ($n = 6$) including the originally analyzed samples. As a result, no dramatic alteration of global transcriptome expression compared to the previous result ($n = 2$) was observed which is also recognizable from the MDS plot (**Supplementary Fig. 2**) although the biological replicates were added. Especially, we found that changes of the transcriptomes of interest that might be involved in the cellular endeavors to resist the acidic milieu (i.e., upregulation of H^+ -pump, inactivation of CCM, modification of the cellular composition, vitalization of protein recovery processes, and promotion of ATP synthesis), were consistently maintained when the number of the replications was increased (**Fig. 1d**). The Gene Ontology enrichment assay with the added sample also showed the similar result with the previous one (**Supplementary Table 1**). Thus, we could arrive at the same conclusion from the newly performed RNA-Seq that overexpression of PMA may be required to enhance the acidic tolerance while securing improved statistical meaningfulness. Plus, we additionally conducted qPCR with the algal PMA genes (i.e., PMA2 and PMA3) which verified the reproducibility of the RNA-Seq (**Supplementary Figs. 6a–6b**). Please refer to the revised manuscript **lines 94–95 and 213–218**, where the related discussions are reflected.

4. Lines 149-154: It is unlikely that downregulation of CCM genes is “due to reducing further endomembrane acidification,” unless this were the signal for regulation of CCM genes, for which there is no evidence that I am aware of. Most CCM genes are expressed in low (ambient) CO₂ but not high CO₂, because they are controlled by the regulator CCM1/CIA5.

- *We are appreciative of your perceptive point. The authors agree that a master gene, CCM1/CIA5, overall control the CCM, including CO₂ transport into the cell. We thus have entirely rephrased the sentence to emphasize what the authors originally intended to discuss while considering your point. Meanwhile, even though intracellular acidification may not serve as a direct signal for regulating the CCM genes, it is worth noting that there is an obvious relevance between pH homeostasis (related to the intracellular acidification) and inorganic carbon (CO₂ and HCO₃⁻) transport which is tightly regulated by the algal CCM system (Yamano et al., Proc. Natl. Acad. Sci. U.S.A., 2015). Please refer to the revised manuscript **lines 166–171**, where the related discussions are reflected.*
- *Regarding this issue, we could discuss one more interesting thing. Intriguingly, according to a recent proteomic study, the algal native PMAs (i.e., PMA2 (Cre10.g459200) and PMA3 (Cre03.g164600)) are revealed to co-function with the core components of the CCM (Mackinder et al., Cell, 2017) that also implies the possible link between the H⁺ extrusion (mainly governed by the PMAs) and CCM. Considering this, the co-functionality of the PMAs with the CCM could be a help in explaining the unexpected PMA expression in the model microalga, *C. reinhardtii*, under high CO₂ conditions. However, further studies seem to be required to clearly identify the precise mechanisms. Please*

refer to the revised manuscript **line 517–531**, where the related discussion is reflected.

- Figure legends should specify the number of biological replicates for each experiment. Preferably, this should be at least three in each experiment.

· *We thank for your comment. We have clarified the number of experiments and meaning of error bars (standard deviation) in the corresponding figure legends and the "Statistical analysis" in the "Methods" part. Please refer to the revised manuscript **lines 818–826** (the "Statistical analysis" section) and **1046–1119** (the figure legends) where the related revisions are reflected.*

- How does the photoautotrophic productivity of the engineered lines (and wild type) in extremely high CO₂ (20% CO₂ or flue gas) compare with productivity in ambient CO₂?

Compiled graph of Fig. 3c and Supplementary Fig. 11c. Optical densities measured at 800 nm were converted into dry cell weights via the correlation presented in Supplementary Fig. 12. Solid and dashed line indicate final time points to estimate the algal biomass productivities under the ambient CO₂ and high CO₂ culture conditions (until their maximum growths), respectively. Details are distinguished by the strain-specific colors.

- Thank you for your question. First, in view of accumulative biomass productions (in g L^{-1} as shown in **the attached graph** above; Please find the corresponding raw data in **Source Data file**) which are achievable from each batch culture, the wild type showed 0.66-fold decreased production (0.925 g L^{-1} , by Day 6) under 20% CO_2 milieu compared with that under ambient CO_2 conditions (2.708 g L^{-1} , by Day 17) while the engineered cell lines, PMA44Cter-V4 and PMA44Cter-V10, exhibited rather 1.12-fold (3.079 g L^{-1}) and 1.09-fold (2.905 g L^{-1}) increased biomass production (by Day 7), respectively, compared to those (2.757 and 2.663 g L^{-1}) under the ambient control conditions (by Day 17) during shorter culture periods. These improvements will undoubtedly contribute to enhancing the practicality of algae as a CO_2 sequestrator from industrial flue gas streams. In addition, given that cell density is one of the most important factors that affect the cost and energy requirements in algae-based products production (Zheng et al., Appl. Energy, 2013), the upgraded tolerance and resultingly increased final biomass concentration may also be a great help to strengthen the economic competitiveness of various CO_2 -originated bioproducts.
- Meanwhile, in terms of kinetic aspects (in $\text{g L}^{-1} \text{ d}^{-1}$; for periods during which the cells reached their maximum concentrations), the engineered algal cell lines, PMA44Cter-V4 and PMA44Cter-V10, showed 2.77-fold ($0.513 \text{ g L}^{-1} \text{ d}^{-1}$) and 2.62-fold ($0.484 \text{ g L}^{-1} \text{ d}^{-1}$) improved biomass production rates under the elevated CO_2 conditions (20%) compared to those (0.172 and $0.166 \text{ g L}^{-1} \text{ d}^{-1}$) (by Day 7) under ambient CO_2 conditions (by Day 17), respectively, thanks to the enhanced CO_2 tolerance. On the contrary, the wild type exhibited only 1.09-fold (9.33%) increased productivity ($0.185 \text{ g L}^{-1} \text{ d}^{-1}$) under the high CO_2 concentration (by Day 6) compared to its production rate ($0.169 \text{ g L}^{-1} \text{ d}^{-1}$)

under atmospheric CO₂ conditions (by Day 17). Considering that relatively high CO₂ conditions (5%) causes over 2-fold improved growth rate in the wild-type strain (0.340 g L⁻¹ d⁻¹) compared to the that under the control (no stress) conditions (**Supplementary Fig. 1**), it could be recognized that 20% CO₂ conditions explicitly inhibits the growth. The lowered productivity (and resultingly worsened CO₂ fixation rate) under the high CO₂ provision inevitably leads to the generation of unfixed CO₂, thereby dramatically reducing CO₂ removal efficiency of the biological system (Chiu et al., *Bioresour. Technol.*, 2008). This necessitates strategies to improving algal CO₂ tolerance, like the technology presented in this work, to harness microalgae in actual CO₂ reduction application. Please refer to the revised manuscript **lines 370–406 and Supplementary Figs. 11c–11d**, where the related discussions are written.

Regarding the proof-of-concept outdoor cultivation with actual flue gas stream, we only checked biomass productivities and several performances of the wild type and a transgenic algal strain, PMA4ΔCter-V4, respectively, with providing the coal-fired toxic flue gas while focusing on proving improved CO₂ tolerance. In other words, we did not inspect their productivities with ambient CO₂ supply. Thus, unfortunately, it is hardly possible to quantitatively compare the strains' biomass productivities against the noxious flue gas supply conditions to those under atmospheric CO₂ provision conditions, based on experimental results. However, given the lab-scale results that the transgenic microalgal cell did not show any compensated growth under atmospheric CO₂ conditions (**Fig. 3c and Supplementary Fig, 11c**), it is predicable that the mutant's growth rate under the toxic flue gas supplement conditions would be much higher than that under ambient CO₂ feeding conditions in the outdoor

due to the enhanced tolerance to high CO₂ and other acidic components (e.g., SO_x and NO_x).

7. Data availability: All RNA-seq reads should be made available through a publicly accessible repository (e.g., NCBI GEO), and an accession number should be provided.

- *We appreciate your suggestion. According to your comment, all RNA-Seq reads were uploaded in a public repository, the NCBI Sequence Read Archive (SRA) as the SRA accession numbers of SRR14160260, SRR14160261, SRR15243031, SRR15243032, SRR15243035, and SRR15243036 (six replicates of ambient conditions-grown cells; atmospheric pH 7.0), SRR14160258, SRR14160259, SRR15243027, SRR15243028, SRR15243029, and SRR15243030 (six replicates of low pH conditions-grown cells; pH 5.0), and SRR14160256, SRR14160257, SRR15243025, SRR15243026, SRR15243033, and SRR15243034 (six replicates of high CO₂ conditions-grown cells; 20% CO₂-enriched air) as indicated in the "Data availability" section (the revised manuscript **lines 829–838**).*

8. Overall, the manuscript needs extensive copyediting for English grammar and readability. I have noted a few instances in the minor comments below, but there are many more. Minor comments:

- *We thank for your comment. We have edited the Manuscript and Supplementary Information thoroughly using Springer Nature Author Services for improving English grammar and readability of this paper (the certificate is attached to the Cover Letter). Including the instances noted by you (please find*

the revised parts from the responses to the individual comments below), several expressions have been corrected while retaining the meanings that the authors intended to convey to the readers.

9. Abstract, line 22: “prone to CO₂” does not make sense

- Thank you for the detailed point. We have erased “prone to CO₂.” Instead of that, we have rewritten the part as “susceptible to CO₂” to clarify the intended meaning. Please refer to the revised manuscript **line 22** where the change is reflected.*

10. Abstract, line 26: “first time demonstrated” sounds awkward

- We appreciated your comment. We have changed the expression from “first time demonstrated” to “for the first time demonstrated” to improve the readability of the abstract. Please refer to the revised manuscript **line 26** where the revision is reflected.*

11. Abstract, line 30: “practical toxic” does not make sense

- We thank for the point. We have deleted "practical" for better readability. Please refer to the revised manuscript **line 29** where the change is reflected.*

12. Line 45: should specify “high CO₂”

- The authors thank for your suggestion. According to your comment, the authors have specified the CO₂ conditions by rewriting the expression from*

*"towards CO₂" to "to high concentration of CO₂" throughout the manuscript.
Please find the sentence in the revised manuscript **lines 44 and 162**.*

13. Line 69: do you mean “too low” instead of “low enough”?

- *Thank you for the detailed point. We totally agree that the previous expression sounds awkward. Therefore, we have revised the expression according to your comment. Please refer to the revised manuscript **line 69** where the change is reflected.*

14. Line 81: delete “lucrative”

- *We appreciated your suggestion. We have changed the word "lucrative" from "profitable" in the sentence. Please find the changed sentence in the revised manuscript **line 81**.*

15. Fig. 1c: Can this figure be replotted so that the areas in the Venn diagrams are proportional to the number of genes?

- *Thank you for your advice. We have replotted the areas of the Venn diagrams to be proportional to the number of the corresponding DEG genes. Please refer to the revised manuscript **Fig. 1c** where the revision is reflected.*

16. Line 195: specify that PMA3 is “the gene of interest”

- *The authors thank for the detailed comment. According to your suggestion, we have specified the name of the genes (i.e., the native *C. reinhardtii* PMA genes)*

*while appending several additional discussions regarding the PMAs and their expressions. Please refer to the revised manuscript **line 209** where the revision is reflected.*

17. Line 258: delete “vivid”

- *We appreciated your suggestion. We have deleted the word "vivid" for better readability. Please refer to the revised manuscript **line 288** where the change is reflected.*

18. Supplementary Fig. 5: legend should state that WT cells are shown in panel (b) instead of panel (a)

- *The authors really appreciate your detailed review on the work and comment on it. We have corrected the previous typo by changing the legend from (b) to (a). Please refer to the revised Supplementary Information **line 366** (in the figure legend of **Supplementary Fig. 9**).*

19. Line 415: change “practical” to “actual”

- *We thank for the suggestion. According to your comment, we have changed the expression from "practical" to "actual." Please find the change in the revised manuscript **line 474**.*

References

1. *Kawasaki, S. et al., Gene expression profiles during the initial phase of salt stress in rice. Plant cell **13**, 889–905 (2001)*

2. Jaeger, D. et al. *Time-resolved transcriptome analysis and lipid pathway reconstruction of the oleaginous green microalga *Monoraphidium neglectum* reveal a model for triacylglycerol and lipid hyperaccumulation.* *Biotechnol. Biofuels* **10**, 197 (2017)
3. Zienkiewicz, A. et al. *The microalgal *Nannochloropsis* during transition from quiescence to autotrophy in response to nitrogen availability.* *Plant Physiol.* **182**, 819–839 (2020)
4. Yamano, T., Sato, E., Iguchi, H., Fukuda, Y., Fukuzawa, H. *Characterization of cooperative bicarbonate uptake into chloroplast stroma in the green alga *Chlamydomonas reinhardtii*.* *Proc. Natl. Acad. Sci. U.S.A.* **112**, 7315–7320 2015
5. Mackinder, L. C. M., et al. *A Spatial Interactome Reveals the Protein Organization of the Algal CO₂-Concentrating Mechanism.* *Cell* **171**, 133–147 (2017).
6. Zhang, Y. et al. *High-density fed-batch culture of a thermotolerant microalga *Chlorella sorokiniana* for biofuel production.* *Appl. Energy* **108**, 281–287 (2013).
7. Chiu, S.-Y. et al. *Reduction of CO₂ by a high-density culture of *Chlorella* sp. in a semicontinuous photobioreactor.* *Bioresour. Technol.* **99**, 3389–3396 (2008).

Final Remarks of the Authors to the Reviewers

We highly appreciate your insightful and constructive comments in our manuscript. All of your logical criticisms and questions undoubtedly help improve the quality of the study and updated manuscript at the same time. We sincerely did our best on revising the manuscript according to your guidelines. We hope that you are satisfied with our answers and revisions. We also wish that the current form of the manuscript meets high standard suggested by Nature Communications and is acceptable for the publication. Once again, thank you for your kind comments and instructions.

REVIEWERS' COMMENTS

Reviewer #1 (Remarks to the Author):

I am positively impressed by the amendments introduced by the authors in the revised version of their manuscript. In my opinion, this manuscript is now mature for publishing.

Reviewer #2 (Remarks to the Author):

The authors answered all the concerns sufficiently. I have no further comments.

Reviewer #3 (Remarks to the Author):

This revised manuscript has been improved in response to the three reviewers' comments. My major concern about the RNA-seq analysis has been addressed satisfactorily by the addition of more replicates. The grammar and readability have been improved, but the manuscript would still benefit from careful copyediting.

Point-by-Point Response to the Reviewers' Comments of

Manuscript NCOMMS-21-11380B

Title: Augmented CO₂ tolerance by expressing a single H⁺-pump enables microalgal valorization of industrial flue gas

We would like to express our thanks for your time and efforts in reviewing our manuscript. We are very appreciative of your great interest in our manuscript and truly pleased to hear that the reviewers are satisfied with our revised manuscript. Once again, we thank you for your invaluable comments, questions, and recommendations derived from the reviewer's thorough analysis, which undoubtedly helped improve the quality of the study.

Comments from Reviewer #1

I am positively impressed by the amendments introduced by the authors in the revised version of their manuscript. In my opinion, this manuscript is now mature for publishing.

- *The authors are genuinely pleased to be informed that you are positively impressed by the revised version of our manuscript. Your invaluable comments will always be an important consideration for our future works as well. Once again, thank you for your great efforts in reviewing this work.*

Comments from Reviewer #2

The authors answered all the concerns sufficiently. I have no further comments.

- *We are very delighted that all the concerns were sufficiently addressed. Your insightful recommendations will be a salient guideline in designing and performing our future research works. Once again, the authors thank you for your great efforts in reviewing this article.*

Comments from Reviewer #3

This revised manuscript has been improved in response to the three reviewer's comments. My major concern about the RNA-seq analysis has been addressed satisfactorily by the addition of more replicates. The grammar and readability have been improved, but the manuscript would still benefit from careful copyediting.

- *We are happy to hear that you are satisfied with our revised work. Your constructive comments will undoubtedly be of great help for designing and conducting our future research works. We will sincerely do our in copyediting process best to improve the manuscript's quality as well. Once again, we highly appreciate your great efforts in reviewing this article.*